

# Comparing short term intensity fluctuations and an Eyewall replacement cycle in Hurricane Irma (2017) during a period of rapid intensification

William Torgerson[1], Juliane Schwendike[1], Andrew Ross[1], and Chris Short[2]

[1]School of Earth and Environment, University of Leeds, LS29JT, Leeds, UK
[2]MetOffice

**Correspondence:** William Torgerson (ee16wst@leeds.ac.uk)

**Abstract.**

An eyewall replacement cycle that occured after a period of rapid intensification of Hurricane Irma (2017) between 07 September and 08 September was investigated in a detailed modelling study using Met Office Unified Model (MetUM) convection permitting ensemble forecasts. The eyewall replacement cycle was then compared to intensity fluctuations that occurred during the period of rapid intensification between 04 September and 06 September. Both the short term fluctuations and eyewall replacement cycle involved an initial dynamical response to localized convection from inner rainbands in the former case and outer rainbands in the later case. One key difference between the intensity fluctuations and the eyewall replacement cycle is that, in the case of the intensity fluctuations, the small radial distance between the eyewall and the inner rainbands meant that a moat region did not form and hence the intensity fluctuations did not lead to an eyewall replacement.

## 1 Introduction

The intensity of a newly formed tropical cyclone (TC) will typically increase over time, with mean sea level pressure falling and tangential wind speed increasing. The period from cyclogenesis to peak intensity will likely take several days but the intensification is rarely continuously monotonic. Instead, there will typically be short periods of weakening prior to the storm reaching its peak intensity. Intensity fluctuations can manifest themselves in many different forms. One example of an intensity fluctuation, which occurs during storms undergoing rapid intensification, is a vacillation cycle (Nguyen et al., 2011) where the potential vorticity (PV) structure of the TC changes from a symmetric to an asymmetric configuration over the course of around 5 hours with a corresponding change in intensity. Another example of an intensity fluctuation includes diurnal pulsing of convection (Dunion et al., 2014) which can lead to intensity changes over a 24 hour period.

An eyewall replacement cycle is a common form of intensity fluctuation in intense and mature TCs. Eyewall replacement cycles involve the replacement of a storm's eyewall with a second developing outer eyewall. During the eyewall replacement cycle the radius of potential hazards tends to increase such as storm surge through the rapid expansion of the storm's wind field. The secondary eyewall formation (SEF) has several possible mechanisms which include: vortex Rossby wave (VRW) activity (e.g. Ruan et al., 2014), beta skirt axisymmetrisation (e.g. Terwey and Montgomery, 2008), and unbalanced dynamics



(e.g. Wang et al., 2016). In all of the above mechanisms dynamical adjustment to outer rainband activity also plays a role with
outer rainbands being a ubiquitous feature prior to SEF. There are also mechanisms that involve asymmetric features such as a
descending inflow jet present in the downshear–left quadrant of Hurricane Earl (2010) (Didlake et al., 2018) which was thought
to play a role in the transition of the rain–band to a secondary eyewall.

Another type of intensity fluctuation has been observed in the case of Hurricane Irma (2017) during its second period of rapid
intensification. Fischer et al. (2020) used observational data to identify two periods of weakening during rapid intensification
where the radius of maximum azimuthally averaged tangential wind (RMW) suddenly increased. The two periods of weakening
were hypothesised to have different causes but were both linked to lower tropospheric convergence and vortex Rossby wave
activity. These intensity fluctuations identified by Fischer et al. (2020) had relatively small intensity changes compared to most
eyewall replacement cycles, but did involve an expansion of the RMW which, as in the case of a full eyewall replacement
cycle, can increase the radius of gale force winds and increase the probability of a storm surge. A modelling study was
undertaken in Torgerson et al. (2023) to determine the characteristics and cause of these intensity fluctuations. It was found
that the intensity fluctuations were linked to the development of localized regions of convection within the inner rainbands. The
dynamical consequence of the inner rainband convection was an acceleration of outflow above the boundary layer leading to a
weakening of the tangential wind within the eyewall. One possible explanation for the acceleration of the outflow jet involves
an unbalanced process within the boundary layer where an increasing agradient wind outside the RMW also acts to increase
the outflow jet above the boundary layer within the eyewall.

The aim of this paper is to understand the eyewall replacement cycle that occurred between 07 and 08 September 2017 in
Hurricane Irma and compare it to the intensity fluctuations that occurred between 04 September and 05 September 2017 with a
view to understanding the differences and similarities between these two different forms of intensity fluctuations. Furthermore,
the paper attempts to understand whether these two different forms of intensity fluctuations are distinct and unrelated or form
part of a continuous spectrum which includes eyewall replacement cycles at one extreme and short term intensity fluctuations
occuring during rapid intensification at the other.

The paper is organised as follows: Section 2 describes the methodology used in the analysis both in terms of the numerical
model setup and the observational data. Section 3.1 compares observational data from the eyewall replacement cycle on 07–08
September to the model output to check that the MetUM is capable of replicating the eyewall replacement cycle which is then
analysed in detail in section 3.2 and compared to established mechanisms for SEF. Once the cause and dynamical processes
involved in the eyewall replacement cycle have been established section 3.2, compares these processes with the intensity
fluctuations to establish similarities and differences between the two forms of fluctuations. An attempt to understand whether
or not these two different types of fluctuations are manifestations of similar dynamics is given in section 3.3. An in–depth
discussion on the causes of the eyewall replacement cycle and its differences to the intensity fluctuation is then presented in
section 4. Conclusions are given in section 5.



## 2 Methodology

### 2.1 Microwave imagery

Morphed integrated microwave imagery (MIMIC) (Wimmers and Velden, 2007) provides useful structural information about the TC which can be compared directly to the numerical model output. MIMIC blends microwave satellite imagery from 85GHz sources to create an image that's brightness is inversely correlated to the presence of deep, rain bearing, convection. The blending algorithm yields an output every 15 minutes and therefore provides regular information on the convective structure of the TC.

### 2.2 Numerical model

For the purposes of comparing an eyewall replacement cycle with short term intensity fluctuations two convection-permitting ensemble Met Office Unified Model (MetUM) simulations initiated on 03 September 00 UTC (covering the short term intensity fluctuations) and 05 September 12 UTC (covering the eyewall replacement cycle) were run for 96 hours. The ensembles are one way nested within the global MetOffice ensemble prediction system (MOGREPS-G).

The MetUM solves nonhydrostatic, fully compressible, deep-atmosphere equations of motion using a semi-implicit, semi-Langangian scheme. It has a wide set of paramtisation schemes for various physical processes. The MetUM in this case is run in the Regional Atmosphere and Land – Version 1 (RAL1-T) configuration presented in Bush et al. (2020). The horizontal grid spacing is .04 deg (approximately 4.4km) in both directions, and there are 80 vertical levels with a horizontal lid 38.5 km above sea level with a model time step of 75 s. More comprehensive details of the model setup are available in section 3.3 of Torgerson et al. (2023) which uses an identical setup for the same storm.

Many of the results in this manuscript are azimuthally averaged. In order to do this a robust TC centre finding method is required. In this case the method used maximises the surface tangential wind in an annulus surrounding the prospective centre which is iteratively found using a simplex algorithm. The exact details of this method are described in section 3.5 of Torgerson et al. (2023) which uses an identical method.

### 2.3 Tangential wind budget

The change in intensity of a TC can be examined in terms of the tangential wind budget which describes contributions to the mean tangential wind tendency from radial and vertical advection of absolute angular momentum, which can be further split up into mean and eddy contributions. A form of the tangential wind budget based on Persing et al. (2013) is:

$$\frac{\partial \overline{v}}{\partial t} = -\overline{u}\,\overline{(f + \zeta)} - \overline{w}\frac{\partial \overline{v}}{\partial z} - \overline{(u'\zeta')} - \overline{\left(w'\frac{\partial v'}{\partial z}\right)} + F, \tag{1}$$

where $v$ is the tangential wind, $u$ is the radial wind, $w$ is the vertical velocity, $f$ is the Coriolis parameter, and $\zeta$ is the relative vorticity. Overbars represent azimuthal averages of these terms while primes represent perturbations from the azimuthal average. The terms on the right hand side of the equation from left to right are: mean radial vorticity flux, mean vertical





advection of absolute angular momentum, eddy radial vorticity flux and vertical eddy advection of absolute angular momentum. The final term, $F$, represents sub–grid frictional contributions to the budget which are negligible outside of the boundary layer.

## 2.4 Calculation of equivalent potential temperature ($\theta_e$)

Equivalent potential temperature ($\theta_e$) provides useful information about the thermodynamic structure of the storm and functions
as a sensible proxy for entropy. $\theta_e$ is used to analyse the effect of the secondary eyewall on the inner eyewall and understand why the inner eyewall dissipates from a thermodynamic perspective. The MetUM does not output the $\theta_e$ field; therefore, the variable had to be calculated using a suitable approximation. Full details of how this calculation was carried out are available in Appendix A.

## 3 Results

### 3.1 Model evaluation

Hurricane Irma (2017) underwent a full eyewall replacement cycle between 07 and 08 September. In order to understand the cause of the eyewall replacement cycle model ensemble simulations were conducted and compared to microwave data. The ensemble member that best matched observations in terms of structure and development was picked for detailed analysis of the eyewall replacement cycle (ensemble member 10 from 05 September 12:00 UTC simulation). The SEF event happened earlier
in reality (10:15 UTC on 07 September compared to 05:00 UTC on 08 September). Figure 1 shows the layer (4062–7038 m) averaged vertical velocity in the mid troposphere (where convection is usually at its strongest) from the model output on the left with the concomitant microwave imagery on the right. The T+0.0 h mark has been taken to be the approximate time of SEF occurrence.

Figure 1a,b shows that, prior to SEF, Irma develops a strong single outer rainband that spirals out from the eyewall. The
105 structure of this strong outer rainband is similar between the model output and the microwave imagery (albeit with the rainband at a different angle to the TC centre, most prominent to the south east in the model output and most prominent to the west in the microwave imagery). At smaller radii close to the eyewall the rainband presents as a continuous zone of convection whilst at greater radii the convection is disparate and consists of individual thunderstorms. However, the strong outer rainband feature does appear earlier, relative to the time when SEF occurs, compared to the model output (9 hours prior to SEF in the microwave
imagery compared to 3.5 hours prior in the model output).

The structure of the TC around the time of SEF is shown in Fig. 1c,d. The prominent outer rainband in 1a,b gives way to multiple outer rainband features which increasingly become apparent at all azimuthal angles and eventually begin to form a ring structure. Down–draughts associated with the many banded features also tend to axisymmetrise and form the beginnings of a moat region between the eyewall and the SEF region. Although the process takes longer in the microwave imagery compared
to the model simulation, the transition from a single dominant outer rainband to multiple rainbands to a ring–like structure is similar in the model output and the microwave observation.





The convection within the secondary eyewall region takes a few hours to organise both in the model and the microwave imagery as shown in Fig. 1e,f. Both the moat region and the secondary eyewall become stronger and more coherent rings of positive and negative vertical velocity respectively during this period. The inner eyewall remains prominent, though in both the model simulation and microwave imagery the ring shape is broken on one side.

The transition between two concurrent eyewalls (Fig. 1e,f) to the complete replacement of the inner eyewall (Fig. 1g,h) takes a little over ten hours in both the microwave imagery and model output. Before this replacement occurs, both the inner and outer eyewalls in both the microwave imagery and model output become less well organised and more asymmetrical with ragged and disorganised convection. The inner eyewall then dissipates and the secondary eyewall reorganises into a coherent ring structure once again. Some remnant convection from the inner eyewall is still visible in Fig. 1g and Fig. 1h and these remnants tend to last at least another 24 hours. This convection associated with the decaying inner eyewall slowly moves inwards into the eye and becomes weaker while it does so.

In general the full eyewall replacement cycle of Irma is captured well by the model simulation with the key structural changes taking approximately the same time. The key difference between the model simulation and the microwave data is that, in reality, it takes longer for the single strong outer rainband to axisymmetrise into a coherent ring of convection.

## 3.2 Secondary eyewall formation (SEF) dynamics

The aim of this section is to understand the cause of the SEF that occurred in Hurricane Irma by analysing the ensemble simulation that captures the eyewall replacement best. Plausible mechanisms from the literature are investigated and compared to the model output to determine the role these mechanisms (if any) played in the SEF. Specifically a detailed investigation has been undertaken into the role of three plausible mechanisms.

1. Vortex Rossby waves (e.g. Ruan et al., 2014), whereby eddy kinetic energy accumulates from outward propagating (mostly wavenumber–2) anomalies outside the RMW providing favourable conditions for SEF.

2. The beta skirt axisymmetrisation (BSA) mechanism (e.g. Terwey and Montgomery, 2008), which is a consequence of two–dimensional fluid dynamics where a weak negative vorticity gradient encourages the axisymmetrisation of turbulent eddies leading to the formation of a jet which can spin up into a secondary eyewall.

3. Unbalanced dynamics within the boundary layer (e.g. Wang et al., 2016), whereby the broadening of the tangential wind above the boundary layer (e.g. due to the dynamical adjustment from the outer rainband activity) leads to increased frictionally induced inflow within the boundary layer, in turn promoting a supergradient wind and an outward agradient force lofting air out of the boundary layer in the SEF region.

### 3.2.1 Vortex Rossby waves

In order to investigate the role of VRWs, wavenumber–2 decomposed PV is used as a proxy to track the VRWs and their radial motion. The VRW activity can then be compared to the time and radial location of the SEF to give some insight into whether or not the VRWs play a role in the eyewall replacement cycle.



Figure 2 shows the wavenumber–2 decomposed PV with a phase angle of zero degrees (eastern azimuth), i.e. values of this
variable only directly east of the storm centre (this choice is arbitrary and picking a different direction only changes the phase
of the anomaly). A VRW event starts in the eyewall at T+71 h (positive wavenumber–2 PV anomaly also denoted by the blue
dashed line) and propagates outwards to the stagnation radius by T+73.5 h. The VRW is convectively coupled and associated
with local ascent (region of positive wavenumber–2 PV anomoly is coincident with high vertical velocity). The azimuthal phase
velocity of the VRW (not shown) was also found to be consistent with the dispersion relation for VRWs (see Montgomery and
Kallenbach, 1997). There is a distinct wavenumber–2 PV anomaly from T+75 h to after SEF (around T+77 h) shown in Fig.
2. However, these PV anomalies did not propagate in a way consistent with the dispersion relation so many not be coherent
VRWs.

In order to determine the effect of the VRW on the tangential wind field, Fig. 3 shows the azimuthally–averaged vertical
velocity and tangential wind acceleration as a function of time. The VRW event induces a large positive tangential acceleration
near the stagnation radius at around T+73 h and also leads to some slight increase in the ascent in the same region. However,
it is notable that SEF does not immediately follow this VRW event and there is a slight tangential wind deceleration between
T+74.5 h and T+75 h.

The mean and eddy contributions to the tangential wind budget are shown in Fig. 4 during the VRW event and around the
SEF. Comparing Fig. 4b to Fig. 4a shows that the VRW event led to a eddy–mean interaction (energy from the VRW transferred
to the mean state) above the boundary layer (around 100 km radius and 2.5 km height) and resulted in a tangential acceleration
around the stagnation radius. By contrast Fig. 4c and d show the dominant contribution during SEF was the mean term. The
VRW event, therefore, likely contributed to the tangential acceleration near the stagnation radius prior to SEF but is unlikely
to have been the direct cause of it.

### 3.2.2 Beta skirt axisymmetrisation (BSA) mechanism

In order for the BSA mechanism to occur, cumulus convection must be allowed to happen in a region where there is a small
mean negative gradient of vorticity. Outside of the eyewall there exists a region called the 'skirt' where this condition for a
small mean negative vorticity gradient is met. However, there is also a region outside the eyewall called the filamentation zone
where convection is suppressed and is therefore unsuitable for this mechanism. Hence a 'goldilocks' zone must be present at
some radial distance from the storm centre where convection is not suppressed but not so far from the storm centre that it still
occurs within the beta skirt . The first requirement can be quantified using the filamentation time (Rozoff et al., 2006) defined
as:

$$\tau_{fil} = \left( \frac{-\overline{v}}{r} \frac{\partial \overline{v}}{\partial r} \right)^{-1/2}, \tag{2}$$

where $\tau_{fil}$ is the filamentation time, $\overline{v}$ is the azimuthal mean tangential wind, and $r$ is the radial distance from the storm centre.
In order for convection to occur, the filamentation time should be greater than the convective lifetime of a cumulonimbus cell,
usually taken to be 30 minutes or less.





The second requirement is quantified in terms of the effective beta function defined as:

$$\beta = -\frac{\partial \overline{PV}}{\partial r}\frac{\overline{\xi}}{\overline{PV}}, \tag{3}$$

where $\overline{\xi} = f + 2\frac{\overline{v}}{r}$ is the inertial parameter. The beta skirt region is thus defined as anywhere $\beta$ is positive but small [1] compared to its value in the eyewall region just outside of the radius of maximum azimuthally averaged vorticity.

Figure 5 shows how the tangential wind field and the 'goldilocks' zone evolve prior to and after the SEF. A little after T+72 hr (Fig. 5a) there is a region of tangential wind acceleration at around 110 km which is associated with a VRW event. Other than this VRW event the tangential wind acceleration outside of the eyewall remains weak in the hours prior to SEF. Figure 5b shows, however, that the 'goldilocks' zone does expand slightly in the hours prior to SEF with conditions met for BSA in a broad region below 2.5 km at radii higher than 70 km from 75.5 hr onwards. Despite the broad favourable region for the BSA, a distinct increase in the tangential wind acceleration only occurs in this zone from around T+77.5 hr onwards (e.g. Fig. 5c) by which time there is already an organised ring of convection. A weak secondary wind maximum does not develop until around T+80 hr (e.g. Fig. 5d) and is only present, at first, around 4 km. No low–level jet is present, and the rapid strengthening of the tangential wind occurs after SEF has already occurred. It is possible that the development of the beta skirt helps develop the secondary wind maxima but it is evident that the BSA mechanism is not responsible for the SEF in this case.

### 3.2.3 Unbalanced dynamics

The unbalanced mechanism starts with a broadening of the wind field that increases the frictionally induced boundary layer inflow which in turn promotes boundary layer convergence, development of the supergradient wind and further increases the tangential wind as a positive feedback.

Figure 6a shows the development of the agradient wind near the surface along with the tangential and radial wind fields. Figure 6b shows the same but for a height of 552 m (mid boundary layer) where the subgradient wind becomes supergradient in the SEF region. The main change in agradient wind occurs at around T+77 h onwards where at around 60 km to 70 km (in the SEF region) the wind goes from marginally supergradient at 552 m to highly supergradient (Fig. 6b). This increase in agradient wind occurs at approximately the same time as the rapid increase in the surface radial wind at around T+76.5 h at 60 km–70 km (Fig. 6a) and slightly earlier for greater radii. At around 50 km the increase in the surface radial wind and 552 m height agradient wind happens later at around T+79.5 h with the agradient wind increasing prior to the increase in the surface radial wind.

Prior to the development of the agradient wind and increased frictionally induced boundary layer inflow a strengthening of the tangential wind occurs above the boundary layer. Figure 7a shows initially before T+75 h rainbands at around 100-125 km produce a weak radial inflow above the boundary layer which is likely a balanced response to the diabatic heating. This inflow,

---

[1] A non negative beta is not a comprehensive definition of the skirt. As in Terwey and Montgomery (2008) the characteristic scale for axisymmetrisation is comparable to but greater than the root mean square of the eddy pertubation velocity divided by the beta. Hence, the large values of beta in the TC core may be sub–optimal for SEF while consistently positive but small beta defines the skirt region.





in turn, promotes a positive tangential wind tendency in the vicinity of and at slightly greater radii than the rainbands by advecting absolute angular momentum inwards. Just before SEF (Fig. 7b) particularly large tangential wind tendencies are seen above the boundary layer at around 2.5 km and 80 km radius. The outflow just above the boundary layer in the moat region (zone of suppressed convection between the primary and secondary eyewalls) is also enhanced by the rainband leading to increased convergence in the SEF region. By T+81 h (Fig. 7c) coupling has clearly occurred with the boundary layer, with
a maximum in vertical velocity extending from the upper troposphere to the boundary layer. By 81 h–84 h the new eyewall is completely coupled with the boundary layer and the SEF had completed.

In summary, the cause of the SEF is best explained by unbalanced dynamics which involve a boundary layer inflow leading to increasing agradient outward forces that promote convection in the SEF region. The original cause of this boundary layer inflow is likely to be due to the broadening of the tangential wind occurring above the boundary layer which is primarily a
result of a balanced dynamical adjustment to outer–rainband activity. Although the BSA and the VRW mechanisms do not directly cause the SEF they do create conditions more favourable by also contributing to the broadening of the tangential wind field, above the boundary layer, prior to SEF.

### 3.3   Comparison between eyewall replacement cycle and intensity fluctuations

In order to determine whether or not the intensity fluctuations are caused by a similar mechanism as eyewall replacement cycles
both the modelled eyewall replacement cycle discussed in section 3.2 and the intensity fluctuations that occurred a few days prior during rapid intensification were compared to one another. Examining the PV structure, for example, is useful in making progress towards understanding whether barotropic instability as in Torgerson et al. (2023) plays a role in both the eyewall replacement cycles and intensity fluctuations. In cases where a direct comparison between the eyewall replacement cycle and intensity fluctuations are made, the ensemble initiated on 03 September 00UTC is used to represent the fluctuations while the
ensemble initiated on 05 September 12UTC the eyewall replacement cycle.

### 3.3.1   Structural similarities and differences

The differences in low–level PV structure above the boundary layer between the intensity fluctuations and the eyewall replacement cycle, are encapsulated in Fig.8. In the full eyewall replacement cycle from about T+60 h onwards the radial PV structure becomes more ring–like as the storm gradually intensifies. The value of $PV_0/PV_{max}$ reaches a minimum of around 0.45 and
gradually increases as the outer rainbands become more prominent after T+70 h. As the rainbands axisymmetrise there is little change in the PV structure with only a gradual tendency for the radial PV structure to become more ring–like. As discussed in Section 3.2.2 an outer 'skirt' of PV forms outside the eyewall around the time of SEF (Fig. 8a at line b) but during the SEF process the PV structure within the primary eyewall and eye changes little. After the secondary eyewall becomes dominant (Fig. 8a at line c) the $PV_0/PV_{max}$ metric increases rapidly and the PV structure moves from ring–like to monopolar due
to PV being transported into the new, larger eye and becomes almost completely monopolar by T+88.5 h (Fig. 8 a at line d) where the primary eyewall has completely broken down to leave only remnant convection within the eye. A secondary PV





maximum appears from around T+77 h onwards corresponding to the new RMW and gradually extends upwards and outwards until merging with the larger PV column by around 85 h.

The weakening phases in the intensity fluctuations by comparison (Fig. 8b) show an immediate increase in the $\mathrm{PV}_0/\mathrm{PV}_{\max}$
metric at the start of the weakening phase whereas during the eyewall replacement cycle the weakening of the maximum tangential wind and expansion of the RMW precedes the redistribution of PV which only occurs after the secondary eyewall has become stronger than the inner eyewall. The transport of PV into the eye in the eyewall replacement cycle causes a significant change in the PV structure and the TC does not regain its ring–like PV structure in the boundary layer until around 84 h and remains completely monopolar above the boundary layer until the end of the simulation at 96 h. For the intensity fluctuations
the change in PV is temporary. During the weakening phases PV is transported into the eye and there is less diabatically generated PV in the eyewall but the ring–like PV structure quickly reforms during the start of the next strengthening phase. There is still a long term upward tendency in $\mathrm{PV}_0/\mathrm{PV}_{\max}$ after the eyewall replacement cycle has completed with the TC overall developing a more monopolar PV structure with time.

The change in the azimuthal structure of the PV, which gives some indication of how the symmetry of the storm changes
during the fluctuations, is also different in the eyewall replacement cycle compared to the intensity fluctuations. Figure 9 compares the horizontal structure of the PV during the eyewall replacement cycle (Fig. 9a-d) and during the intensity fluctuations (Fig.9e-h). Figure 9a-c shows during the SEF period there is little change in the azimuthal structure in the PV. The PV structure does, however, become more azimuthally asymmetrical after the secondary eyewall becomes dominant with Fig. 9d showing a more elliptical structure. This change in azimuthal PV structure is quite different to the intensity fluctuations where an initially
elliptical PV ring at the start of the first weakening phase (Fig. 9f) rapidly symmetrises into a more circular structure during the middle of the weakening phase (Fig. 9g). In the case of intensity fluctuations, the increase in azimuthal symmetry was attributed to the weakening of wave–2 inner rainband structures (Torgerson et al., 2023) that are not present in the case of the eyewall replacement cycle. The changes in azimuthal PV symmetry during the eyewall replacement cycle are also smaller than the changes in azimuthal PV symmetry during the intensity fluctuations. The maximum standard deviation of PV (not shown)
does not change much during the eyewall replacement cycle (although there is an increase in the standard deviation of PV near the new eyewall after it becomes dominant) whereas during the intensity fluctuations there is a very rapid decrease in the standard deviation of PV during a weakening phase and a rapid increase in the standard deviation of PV during a strengthening phase.

In terms of comparing the convective structures during the SEF and the intensity fluctuations, Fig. 10 shows the secondary
circulation at key times during the SEF and the intensity fluctuations prior to and during W1. Prior to SEF the vertical velocity increases at a radius coinciding with the rainband region which at T+72 h around 130 km. The updraught centre moves radially inward and slightly decreases in height over time and after SEF (Fig. 10d) ends up merging with the boundary layer to form a coherent updraught at all levels. There is already an outflow above the boundary layer associated with the outer rainband in Fig. 10a which develops into a full tropospheric outflow channel by T+75 h (10 b), during SEF (10 c) this new outflow
channel couples to and merges with the boundary layer outflow from the primary eyewall which can be seen by T+80 h in Fig. 10 d. There is a sequence of events where the secondary circulation associated with the outer rainband gradually becomes





more dominant before entirely replacing the secondary circulation of the primary eyewall. The gradual intensification of an outer pillar of convection is not seen in the intensity fluctuations. Comparing the start of W1 (Fig. 10f) with the secondary circulation prior to the start of W1 (Fig. 10 e) shows that the updraught associated with the eyewall has become bifurcated; this bifurcation becomes increasingly more apparent with two separate maxima by T+47.5 h (Fig. 10g). However, unlike in the case of the eyewall replacement cycle, there is no updraught (associated with outer rainbands) outside of the eyewall and above the boundary layer that moves radially inwards over time before merging with the boundary layer updraught associated with the newly forming secondary eyewall. In the intensity fluctuations there is a weak mid–tropospheric secondary outflow visible at T+45.5 h (Fig. 10f) around 100 km radius and 7 km height but this does not subsequently merge to the boundary layer outflow as it does between T+75 h and T+77 h in the eyewall replacement cycle (Fig. 10b-c).

### 3.3.2 Updraft coupling

In order to understand the differences between the eyewall replacement cycle and the intensity fluctuations it is necessary to re–examine the boundary layer and how it couples with the free vortex above in the case of both the eyewall replacement cycle and the intensity fluctuations.

Figure 11 shows how the agradient, radial and tangential wind fields change through the two types of intensity fluctuations. In the case of the eyewall replacement cycle, prior to SEF the agradient wind gradually increases in the top part of the boundary layer between 75 km and 100 km radius particularly where the outer rainbands are active (Fig. 11a,b) during this time the tangential wind increases slightly above the boundary layer (at around 3.5 km height, 80 km radius) which explains the slight increase in the boundary layer inflow and increase in the agradient wind. At around T+77 h (Fig. 11 c) there is a larger increase in the tangential wind above the boundary layer and an increase in the radial inflow. By 80 h (Fig. 11d) the agradient wind has increased at all levels within the boundary layer (particularly around 60 km) and there is a rapid increase in the tangential wind above the boundary layer to form a secondary tangential wind maximum. In contrast, during the intensity fluctuations the increase in the agradient wind occurs much closer to the eye–wall which can be seen particularly between T+47.5 h and T+50 h (Fig. 11g-h) at around 40–50 km radius within the boundary layer. There are however some similarities. The increase in the agradient wind is also associated with an increase in the boundary layer radial inflow albeit at a relatively narrow radius of around 40–60 km and there is an associated tangential acceleration at the same radius. Additionally, there is also an increase in the tangential wind prior to the weakening phase (Fig. 11e) as there is for the eyewall replacement cycle (Fig. 11b).

### 3.3.3 Mass ventilation

As in Torgerson et al. (2023) understanding the ability of the deep convection to ventilate the mass inflow in the boundary layer can offer additional insight into the role of the unbalanced dynamics in both the case of the eyewall replacement cycle and the intensity fluctuations. The ventilation diagnostic developed in Smith et al. (2021) will be examined in both types of fluctuations. Full details about how the ventilation diagnostic is calculated are available in Torgerson et al. (2023) which, in turn, is based on prior work in Smith et al. (2021). The ventilation diagnostic is defined as in equation 5 of Torgerson et al. (2023):





$$\Delta M_{flux}(R_{int}, t) = 2\pi \int_0^{R_{int}} [<\rho w>_{z=Uppertrop} - <\rho w>_{z=BL}] r dr, \tag{4}$$

where $\Delta M_{flux}$ represents the ventilation diagnostic and triangular brackets indicate azimuthally averaged quantities as a function of the integration radius and time. The subscripts z=BL (boundary layer) and z=Uppertrop (upper troposphere) represent the quantities evaluated at 1052 m and 5955 m respectively.

The ventilation diagnostic (a function of radius and time), if positive, describes the situation where the convection, at that
radius, is able to more than ventilate the mass flow from the boundary layer. On the other hand if the ventilation diagnostic is negative then the convection is not strong enough to remove the mass influx from the boundary layer, and consequently there will exist an outflow jet above the boundary layer containing the unventilated mass.

Figure 12 shows the ventilation diagnostic over time as well as the radial inflow at the surface and outflow above the boundary layer for the W1 eyewall replacement cycle Fig.(12a,b) and the intensity fluctuations Fig.(12c,d). In the case of the
320 intensity fluctuation just prior to W1 the ventilation diagnostic within the eyewall decreases from around -1.5×10⁹ kg s⁻¹ to -2.5×10⁹ kg s⁻¹ while outside the eyewall at around 75 km the ventilation index has gone weakly positive. The increasing radial inflow within the eyewall boundary layer is too strong to be ventilated by the eyewall leading to an increasing outflow at around 46 h above the boundary layer just outside of the eyewall. The convection in the inner–rainband area is strong enough to ventilate the (weaker) boundary layer inflow, so this is a more favourable environment for the spin–up to continue. Boundary
layer convergence also increases the vertical velocity in this region which can be seen at both the 1 km and 6 km level by 47 h. The intensification of the convection at a slightly greater radius comes at the expense of the convection at smaller radii (further details of the nature this process are given in Torgerson et al. (2023), hence the further lowering of the ventilation index for radii under 50 km after 47 h.

For the eyewall replacement cycle the process is subtly different. Within the outer rainband region the ventilation index
actually decreases (in contrast to the fluctuations where the ventilation index became more positive prior to W1) as inflow at these larger radii increases. Nevertheless, the ventilation index does not go negative in the centre of this outer rainband convection (around 110 km at 75 h) so the convection is still just able to vent the strengthening boundary layer inflow. Within the inner eyewall the ventilation index remains stable at around -2×10⁹ kg s⁻¹ and later becomes more positive as the inflow starts to weaken during the replacement. There is a distinct decrease in the ventilation index from near neutral to more than
335 -10×10⁹ kg s⁻¹ in the moat region between the two eyewalls (particularly in the half of the moat closest to the outer eyewall). The suppression of the convection within the moat region makes it completely unable to ventilate the strong boundary layer inflow and hence is responsible for the dramatic increase in the outflow in this region above the boundary layer. Crucially, the strongly negative ventilation index within the moat region implies that in the case of the eyewall replacement cycle it is the moat that is responsible for the increase in outflow rather than the inner eyewall.



### 3.3.4 Demise of the inner eyewall

It is useful to compare how the thermodynamic structure changes during the eyewall replacement cycle and the intensity fluctuations to determine any differences and similarities between the two kinds of intensity fluctuations. We know in eyewall replacement cycles the moat region plays an important role in the demise of the primary eyewall (Zhou and Wang, 2011) so it is useful to investigate whether something similar happens in the intensity fluctuations.

Figure 13 shows the $\theta_e$ structure during key stages of the eyewall replacement cycle and intensity fluctuations. In the eyewall replacement cycle the moat is visible especially at around T+77 h (Fig. 13a) as a region of local descent at around 5 km height and 60 km radius and relatively low radial $\theta_e$ gradients. As the secondary eyewall updraught strengthens and becomes dominant over the inner eyewall, the moat region moves radially inwards and is visible as a region of descent and locally low $\theta_e$ at around 40 km by T+83 h (Fig. 13c). By T+86 h (Fig. 13d) the remnants of the moat are still visible as a region of weak descent around 30 km radius and 2.5 km height. The value of $\theta_e$ in the (inward moving) moat is higher than T+77 h by this point, but both the absolute value of $\theta_e$ and the radial $\theta_e$ gradient have not recovered, with both being considerably lower than they were before SEF. At around 60 km, however, the radial $\theta_e$ gradient is substantially higher than it was before SEF which is the approximate radial location of the new eyewall.

By contrast, the intensity fluctuations do not seem to have any lasting impact on the $\theta_e$ structure of the storm. During the middle of the weakening phase (Fig. 13e) the bifurcated ascent in the eyewall is associated with a low radial $\theta_e$ gradient. However, the gradient never goes positive as it does in the moat region during the eyewall replacement cycle (Fig. 13b) and no low $\theta_e$ air is imported into the eye as it is during the eyewall replacement cycle. Prior to SEF (for example at T+72 h, not shown) during the eyewall replacement cycle a parcel moving inwards would experience a large increase in $\theta_e$ from the moat region to the eyewall on the order of a 17–20 K. After SEF (Fig. 13 c) that has declined to 10–14 K.

## 4 Discussion

Several hypotheses were investigated in an attempt to understand the cause of the eyewall replacement cycle. One of those mechanisms was VRWs. It is shown that a major VRW occured at around T+72 hours and that it is convectively coupled (Fig. 2). The VRW's radial, azimuthal and height position also showed agreement with the dispersion relation and there was also evidence of the existence of a stagnation radius. The VRW event did have an impact on the tangential wind field, notably a wave–mean interaction (Fig. 4b) led to an acceleration of the tangential wind around the stagnation radius at around 3 km height. However, any VRW event that occurred after T+72h had less impact on the tangential wind field and also seemed to be less convectively coupled. It is likely that the VRW event plays an important role in broadening the tangential wind field prior to SEF but is unlikely to have been the direct cause of it.

Another mechanism investigated was the BSA. This mechanism can be ruled out as a direct cause of the SEF as a low level jet did not develop prior to the axisymmetrisation of the convection (Fig. 5c) making a wind induced surface heat exchange feedback process impossible. Additionally, when the secondary wind maximum does develop (Fig. 5d) it first appears above the boundary layer at around 3.5 km height. The convective updraughts associated with the new eyewall merge with the boundary



layer hours earlier at around T+77 h (Fig. 10c) implying that the formation of a coherent updraught effected a secondary wind maximum, not the reverse as would be the case in a BSA mechanism. Nevertheless expansion of the beta skirt did occur prior to SEF (especially after the vortex Rossby wave event at T+72 h which also likely played a role in enhancing the outer core vorticity) and tangential wind acceleration did occur in this increasingly expanding 'goldilocks' zone in the beta skirt outside of the filamentation zone (Fig. 5 a-c, Fig. 7 a-b) albeit with not enough of an acceleration to produce a low–level jet.

The most compelling mechanism for the eyewall replacement cycle is an unbalanced feedback process that begins with a broadening of the tangential wind field, largely above the boundary layer, that can be seen in Fig. 7 a-b. The initial cause of this tangential broadening is likely a balanced adjustment to the outer rainband causing increased inflow above the boundary layer which can be seen especially in Fig. 7b at around 80 km radius. The broadening of the tangential wind in the SEF region above the boundary layer precedes a rapid increase in the agradient wind and development of a supergradient wind in the middle of the boundary layer (Fig. 6b) that occurs between 60 km and 100 km radius after 77 h. The rapid development of the supergradient wind in the middle of the boundary layer at 77 h corresponds to the time when the rainbands axisymmetrise into a coherent tower of vertical velocity (Fig. 10c) indicating the main cause of the spontaneous axisymmetrisation of the rainband into an eyewall structure is the outward agradient force within the boundary layer causing air to erupt out of the boundary layer and merge with the ascent associated with the rainband above.

The cause of the increased agradient wind is more difficult to determine. The mechanism to increase the agradient wind described by Tyner et al. (2018) and others is that tangential broadening above the boundary layer leads to increased frictionally induced boundary layer inflow which in turn through the unbalanced spin–up mechanism (Smith et al., 2009) leads to the increased agradient wind at some height within the boundary layer. The problem is, the surface inflow only rapidly increases after 79 h at around 65 km radius (Fig. 6a) by which time the supergradient wind has already developed above (Fig. 6b). At slightly higher radii (70–80 km), though, the surface inflow does increase earlier at around 76.5 h and there is an increase in the boundary layer inflow at 60–90 km radius at 552 m at around 77 h (Fig. 6b). It is plausible that the lack of obvious causality (with increased surface radial wind preceding the increase in supergradient wind) is due to a positive feedback effect. An initial increase in the boundary inflow accelerates the tangential wind (and increases the agradient wind) in the boundary layer which in turn increases the frictionally induced inflow and further increases the supergradient wind. At the surface there is an increase in the tangential wind after T+76.5 h (Fig. 6). Increases in the agradient wind at the 552 m level are modest between T+76 h and T+78 h on the order of 2 $\mathrm{m\,s^{-1}\,h^{-1}}$ compared to 5 $\mathrm{m\,s^{-1}\,h^{-1}}$ afterwards. It is noteworthy that amongst other studies that do claim the unbalanced mechanism plays a role in SEF this apparent absence of causality is observed. For example in Huang et al. (2012) comparing their Fig. 5a with their 5c shows that between 10 September 15 UTC 10 September 21 UTC the radial inflow does not increase (and may even decrease slightly) yet the agradient wind is increasing during this time. Another example is in Fig. 11b from Chen (2018) where the agradient wind increases monotonically from around T+159 h yet the radial inflow only starts to increase at around T+168 h.

It is therefore proposed that one possible interpretation of the results is a sequence of events that start with the development of outer–rainband activity from T+65 h onwards. Balanced dynamical adjustment to the diabatic heating from the rainband leads to increased inflow above the boundary layer which in turn accelerates the tangential wind field. The acceleration of the





tangential wind field is also aided by a VRW event that occurs at around T+72 h. Between T+72 h and T+77 h the convection associated with the rainband continues to organise and move radially inward and the tangential wind above the boundary layer

increases. Eventually the acceleration of the tangential wind above the boundary layer leads to an increased boundary layer radial inflow and concomitant development of the supergradient wind that causes the air to erupt out of the boundary layer to form the coherent secondary vertical velocity maximum. Once this updraught in the SEF region has appeared at all height levels the development of a secondary tangential wind maximum follows a few hours afterwards.

Having established a plausible mechanism for the eyewall replacement cycle, a comparison with the intensity fluctuations

can be undertaken. In Zhang et al. (2017) during the composite for a full eyewall replacement cycle (their Figure 6) the region of ascent associated with the outer rainband gradually descends and moves radially inwards until T+35 h when it merges with the boundary layer. In contrast in their 'partial eyewall replacement cycle' composite (their Figure 11) the region of ascent associated with the rainband does not develop and merge with the boundary layer, instead ascent beginning at around T+35 h just radially outside of the eyewall and above the boundary layer develops into a second pillar of vertical velocity by T+45 h.

The differences between Figure 6 and Figure 11 in Zhang et al. (2017) are reminiscent of the differences between Fig. 10a–d and Fig. 10e–h in this study suggesting that the 'partial eyewall replacement cycle' in Zhang et al. (2017) may be similar to the intensity fluctuations in Irma. An additional similarity is the lack of a coherent tropospheric outflow channel in the 'partial eyewall replacement cycle' case that does appear in the full eyewall replacement cycle case (as it does in this study's eyewall replacement cycle example). However, the intensity fluctuations did occur over a short time interval (around 6 hours) which

was shorter than the 'partial eyewall replacement cycle's in Zhang et al. (2017). A similar phenomenon to the 'partial eyewall replacement cycle' is described in Wang and Tan (2020) as a 'fake SEF' whereby inner rainband activity can drive boundary layer inflow, convergence and an increase in the agradient wind (as is also seen in the intensity fluctuations) but without the positive feedback associated with a 'wind–maximum' pathway. This lack of 'wind–maximum' pathway represents a plausible difference between eyewall replacement cycles and the intensity fluctuations. Fig. 13 confirms that the SEF is associated with a

large tangential acceleration outside of the eyewall and results in a larger change in the agradient wind compared to the intensity fluctuations where the tangential acceleration is short lived and not big enough to result in a secondary wind maximum.

A key difference between intensity fluctuations that may be driven by inner rainband activity compared to outer rainband activity driven eyewall replacement cycles is the development of a moat (region of relatively low equivalent potential temperature between the primary and secondary eyewall) which can be seen in Fig. 13. The difference between an eyewall replacement

cycle involving a large moat compared to a smaller one is seen in Zhou and Wang (2011) in their Figure 11. With the larger moat the inner eyewall is replaced and low entropy air is imported into the eye (similar to Fig. 13a–d) whereas with the smaller moat the low $\theta_e$ reservoir quickly dissipates and the bifurcated eyewall structure reorganises into a coherent pillar of ascent. In the case of the intensity fluctuations there is no moat at all but the radial $\theta_e$ gradient does become temporarily weaker. As such it is proposed that the thermodynamic structural differences in the eyewall replacement cycle and the intensity fluctua-

tions can be explained by the absence of a moat, which in turn is caused by a balanced and unbalanced response to an inner rainband occurring radially inward of the filamentation zone. As in Zhou and Wang (2011) a likely mechanism for the demise of the primary eyewall is the cutting off of the source of high entropy air. As explained in Emanuel (1991), a hurricane can be



considered like a Carnot cycle with the inflow leg being an isothermal expansion with air parcels picking up entropy from the environment; in the case of the eyewall replacement cycle the secondary eyewall reduces the length of this radial inflow to the primary eyewall, with a smaller entropy change and as a result there is less energy available to the inner eyewall during the ascending leg.

A summary of the proposed mechanism for the fluctuations and the eyewall replacement cycle is given in Fig. 14. In Fig. 14a,b the tangential wind field broadens as a consequence of a dynamical response to convection with that convection being the outer rainband in the case of the eyewall replacement cycle and the inner rainbands in the case of the intensity fluctuations. In Fig. 14c,d unbalanced processes at the interface between the boundary layer and the free vortex above cause an intensification of the new convection outside the eyewall. Finally in Fig. 14e,f the new convection has replaced the original convection which represents a reorganisation of the eyewall for the intensity fluctuation and the demise of the inner eyewall in the case of the eyewall replacement cycle.

## 5   Summary and conclusions

The aim of the paper was to understand the processes at work during a full eyewall replacement cycle in Hurricane Irma (2017) on 07 and 08 September and secondly, to compare those processes to the intensity fluctuations that occurred during rapid intensification a few days earlier. It was determined that the eyewall replacement cycle was likely due to boundary layer convergence, of air, that resulted from a balanced adjustment to outer rainband activity. VRWs and the BSA mechanism were found to play ancillary roles but were not the main cause of the SEF. The subsequent demise of the inner eyewall is hypothesised to be caused by the interruption of the supply of warm, moist, high entropy air by the outer eyewall. There were similarities between the eyewall replacement cycles and intensity fluctuations: notably the integral role of the unbalanced spin–up mechanism in both types of fluctuation. However, there are also many differences including the changing PV structure, changing secondary circulation and the changing $\theta_e$ structure. Most significantly, the intensity fluctuations were a consequence of inner rainband activity while eyewall replacement cycles are a consequence of outer rainband activity with the distinct presence of a moat. Key findings from this analysis include the following:

– The SEF is largely due to unbalanced dynamics: boundary layer convergence is promoted through the development of the supergradient wind, a consequence of the broadening tangential wind field above the boundary layer.

– Vortex Rossby waves do not cause SEF in this case, but play a role in broadening the tangential wind field, an important step to SEF. Development of a beta skirt prior to SEF likely also led to more ideal conditions for SEF to occur by promoting tangential wind acceleration in the beta skirt.

– The eyewall replacement cycle was characterised by a ring–like PV structure prior to and during SEF, followed by a transition to a monopolar PV structure after the secondary eyewall became dominant.



- – The eyewall replacement cycle was characterised by an azimuthally symmetric PV structure that became more asymmetric after the secondary eyewall became dominant. Unlike in the intensity fluctuations, an azimuthally asymmetric PV structure was correlated to a more monopolar PV structure.

- – Prior to SEF the convection associated with the outer rainband became increasingly intense and moved gradually inward and downwards before merging with the boundary layer during SEF. The intensity fluctuations, by contrast, had no similar SEF event and behaved more like a reorganisation of the eyewall structure rather than a replacement.

- – The increase in the supergradient wind and associated boundary layer inflow was much more pronounced compared to the increase in the supergradient wind outside of the eyewall in the intensity fluctuations.

- – Prior to SEF a moat region of descent and low equivalent potential temperature developed that eventually ended up in the eye after the replacement. By contrast the thermodynamic structure of the storm during the intensity fluctuations was stable.

It is proposed that the initiating mechanisms behind eyewall replacement cycles and intensity fluctuations are similar, with both beginning with a dynamical response to diabatic heating from convective structures. In the case of the eyewall replacement cycle diabatic heating is associated with outer rainbands far from the eyewall, while in the intensity fluctuations it is associated with inner rainbands emanating from the eyewall. Despite the similar initial cause, the sequence of events is quite different because in the case of the eyewall replacement cycle the SEF occurs far from the eyewall and is separated from it by a moat region, whereas with the intensity fluctuations the process happens at the outer edge of the eyewall itself. As a result an intensity fluctuation is a transient weakening associated with a brief reorganisation of the eyewall, while an eyewall replacement cycle is a permanent substantial structural change to the storm that necessitates the complete destruction of one eyewall at the expense of building another. The key differences between the two types of fluctuations are summarised in Fig. 14.

*Data availability.* Observational data used in this paper is made available online by the Hurricane Research Division and is available at https://www.aoml.noaa.gov/hrd/Storm_pages/irma2017/. The microwave data is made available online by CIMSS at http://tropic.ssec.wisc.edu/real-time/mimtc/2017_11L/web/mainpage.html. The model fields in a 200km box around the storm which are used for the analysis in this paper have been stored and can be made available on request.

**Appendix A: Calculation of equivalent potential temperature ($\theta_e$)**

The following method based on equation 39 in Bolton (1980) was used to calculate an approximation to $\theta_e$:

$$\theta_e = \theta_L \exp\left(\left(\frac{3036}{T_L} - 1.78\right) r(1 + 0.448r)\right), \tag{A1}$$



where $\theta_L$ is the potential temperature at the lifting condensation level, $T_L$ is the temperature at the lifting condensation level, $r$ is the water vapour mixing ratio. In order to determine $\theta_L$ and $T_L$ the expression for the temperature and potential temperature at the lifting condensation level was used from Romps (2017) which gives equations for $\theta_L$ and $T_L$ as:

$$
\begin{aligned}
T_L &= c\left[W_{-1}\left(RH^{\frac{1}{a}}ce^c\right)\right]^{-1}T, \\
a &= \frac{c_{pm}}{R_m} + \frac{c_{vl}-c_{pv}}{R_v}, \\
c &= \frac{-E_{0v}-(c_{vv}-c_{vl})T_{trip}}{aR_vT},
\end{aligned}
\tag{A2}
$$

$$
\theta_L = T_L\left(\frac{p_0}{p\left(\frac{T_L}{T}\right)^{\frac{c_{pm}}{R_m}}}\right)^{R/c_p},
\tag{A3}
$$

where $W_{-1}$ is the negative one branch of the Lambert-W function, $RH$ is the relative humidity, $T$ is the temperature, $c_{pm}$ is the moist specific heat capacity at constant pressure, $R_m$ is the moist specific gas constant, $c_{vl}$ is the specific heat capacity of liquid water, $c_{pv}$ is the moist specific heat capacity at constant volume, $R_v$ is the specific gas constant for water vapour, $E_{0v}$ is the difference in specific internal energy between liquid and solid at the triple point of water, $c_{vv}$ is the specific heat capacity of water vapour at constant volume, $T_{trip}$ is the temperature of the triple point of water, $p$ is the pressure, $p_0$ is the reference

pressure (1000 hPa), $R$ is the dry specific gas constant. All of these quantities are known prognostic variables or constants (see Romps (2017) for values used) except the relative humidity which also has to be calculated. The relative humidity was output at the surface level which was used to check the calculated value was reasonable.

The relative humidity is given by the ratio of the vapour mixing ratio (prognostic variable) to the saturated vapour mixing ratio:

$$
RH = \frac{r}{r_s}.
\tag{A4}
$$

In order to calculate the saturated vapour mixing ratio the following formula is used,

$$
r_s = \frac{Rp_s}{R_vp},
\tag{A5}
$$

where $p_s$ is the saturation vapour pressure. It is important to obtain the maximum possible accuracy for the relative humidity over a sensible temperature range. Consequently the empirical Arden–Buck equation is used (Buck, 1981) to calculate the

saturation vapour pressure which has good accuracy in the meteorologically relevant temperature range (-50 to 30°C):

$$
p_s = \begin{cases} 6.1121\exp\left(\left(18.678-\frac{T}{234.5}\right)\left(\frac{T}{257.14+T}\right)\right), T > 0°c \\ 6.1115\exp\left(\left(23.036-\frac{T}{333.7}\right)\left(\frac{T}{279.82+T}\right)\right), T < 0°c. \end{cases}
\tag{A6}
$$



*Author contributions.* **William Torgerson**: Conceptualization, Formal Analysis, Investigation, Methodology, Software, Writing – Original Draft Preparation. **Juliane Schwendike**: Conceptualization, Supervision, Writing – Review and Editing. Funding acquisition **Andrew Ross**: Conceptualization, Supervision, Writing – Review and Editing. Funding acquisition **Chris Short**: Data Curation, Methodology, Supervision, Writing – Review and Editing.

*Competing interests.* Some authors are members of the editorial board of WCD. The peer-review process was guided by an independent editor, and the authors have also no other competing interests to declare.

*Acknowledgements.* We thank the Cooperative Institute for Meterological Satellite Studies for making microwave imagery available contained in Fig. 1b,d,f,h (available online at http://tropic.ssec.wisc.edu/real-time/mimtc/2017_11L/web/mainpage.html).

This work used Monsoon2, a collaborative High-Performance Computing facility funded by the Met Office and the Natural Environment Research Council. Torgerson was funded by a PhD scholarship from the NERC SPHERES DTP (grant NE/L002574/1) and CASE support from the Met Office. This research was also partially funded by the Met Office Weather and Climate Science for Service Partnership (WCSSP), Southeast Asia, as part of the Newton Fund.





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



## List of Figures





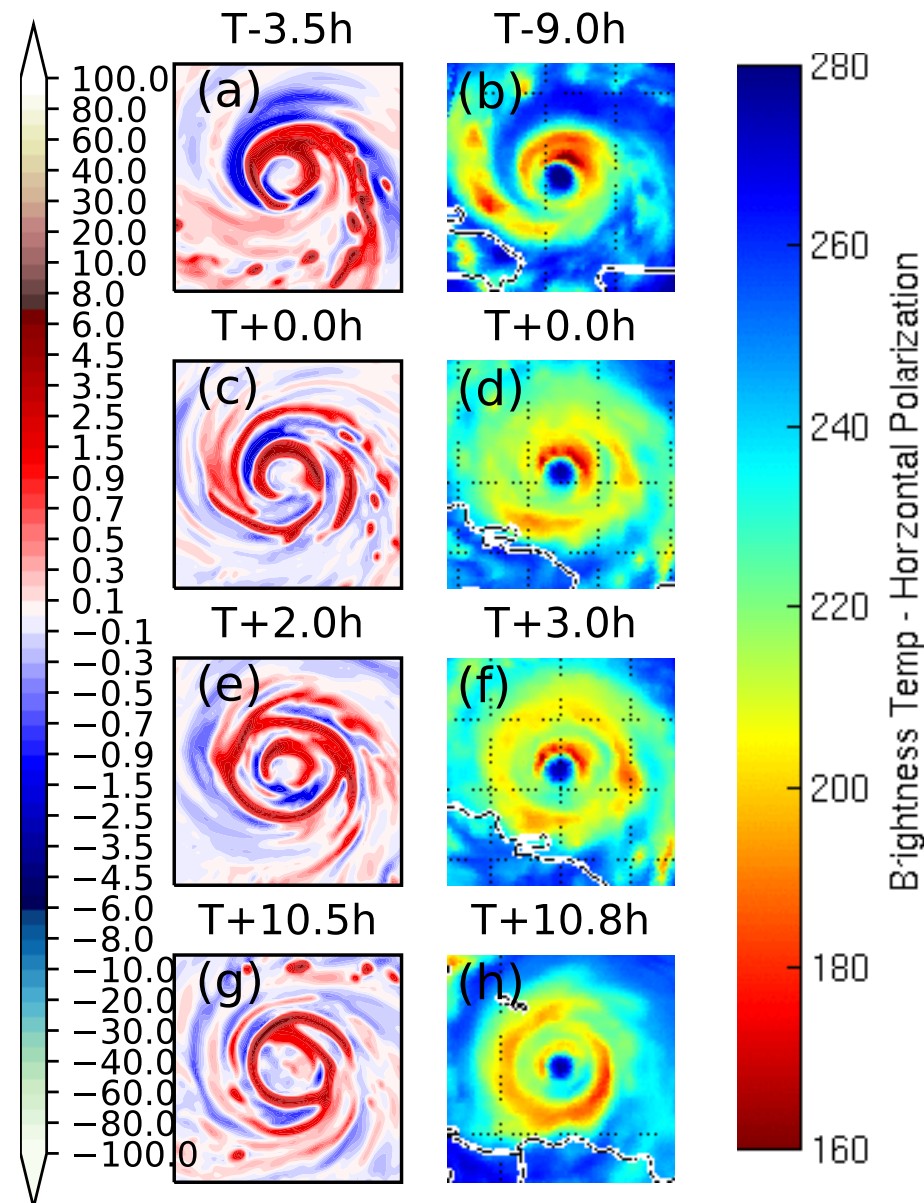

**Figure 1.** (a,c,e,g): Layer averaged (4062–7038 m) simulated vertical velocity ($\mathrm{m\,s^{-1}}$, shading), and (b,d,f,h): microwave imagery at selected times within the eyewall replacement cycle. Times have been chosen such that they are relative to the approximate SEF stage in both the microwave and model data. These stages are: (a,b) single spiral outer–rainband, (c,d) SEF, (e,f) outer eyewall becomes dominant, and (g,h) remnant eye convection.

**Figure 2.** Wavenumber–2 PV for the eastern azimuth (shaded, PVU) at 1532 m. Vertical velocity (black line contours of 0.4, 0.8, 1.6, 3.2 $\mathrm{m\,s^{-1}}$) along the same azimuthal angle and height. Also shown are the RMW for the same height (grey solid line) and 3×RMW as a proxy for the stagnation radius (purple solid line). The blue dashed line shows the trajectory of a hypothetical VRW propagating from the RMW at T+71 h using the VRW dispersion relation.

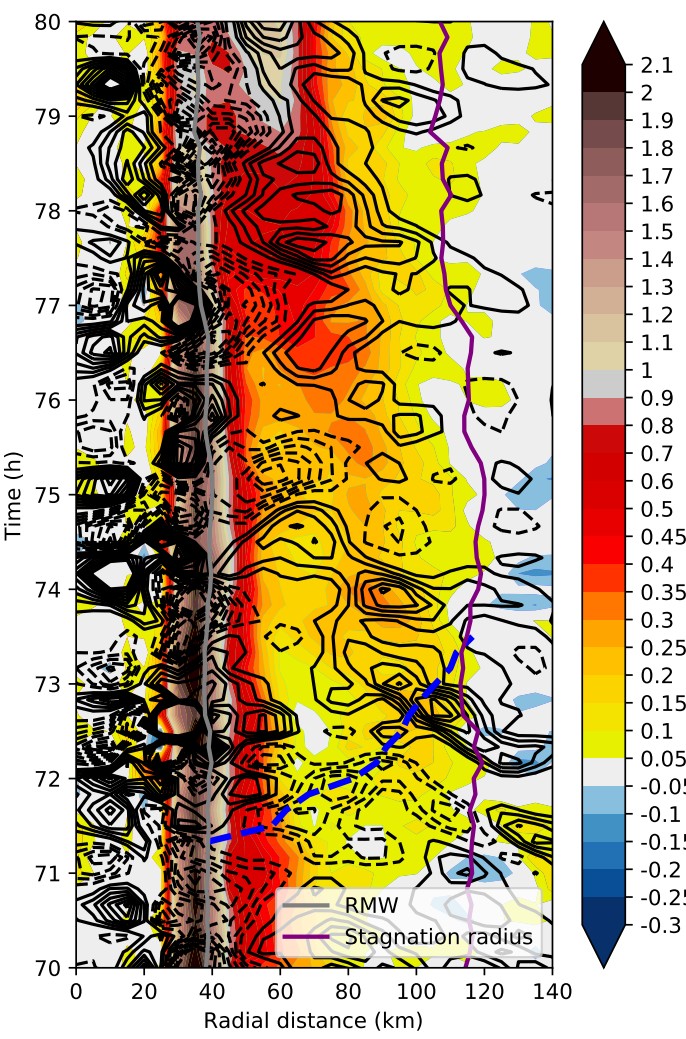

**Figure 3.** Azimuthally averaged vertical velocity at 1532 m (shaded, $\mathrm{m\,s^{-1}}$). Azimuthally averaged tangential wind acceleration (black line contours of 0.5 $\mathrm{m\,s^{-1}}$ intervals). Also shown are the RMW for the same height (thick grey solid line) and 3×RMW as a proxy for the stagnation radius (purple solid line). The dashed blue line shows the radial, time trajectory of a hypothetical VRW propagating from the RMW at T+71 h using the VRW dispersion relation.





**Figure 4.** (a,c) Mean and (b,d) eddy terms (shaded, $\mathrm{m\,s^{-1}\,h^{-1}}$) of the tangential wind budget are shown. The tangential wind (line contours, $10\,\mathrm{m\,s^{-1}}$ intervals) and the RMW (grey dashed line) are also shown.






**Figure 5.** Tangential wind acceleration (shaded, $\mathrm{m\,s^{-1}}$), tangential wind (line contours, $10\ \mathrm{m\,s^{-1}}$ interval) for selected times. Stippling shows regions where the filamentation time $\tau_{fil}$ is less than 30 minutes and cross hatching shows regions where effective beta $\beta$ is negative.





**Figure 6.** Agradient wind (shaded, $\mathrm{m\,s^{-1}}$), tangential wind (black line contours, 5 $\mathrm{m\,s^{-1}}$ interval), and radial wind (orange line contours, 5 $\mathrm{m\,s^{-1}}$ interval, dashed lines imply an inflow) at 12 m height (a), and 552 m height (b).





**Figure 7.** Azimuthally averaged tangential wind tendency (shaded, $\mathrm{m\,s^{-1}}$ over 3 hours), three–hourly azimuthally averaged vertical velocity (black contours, 0.25, 0.5, 1 $\mathrm{m\,s^{-1}}$ intervals and 1 $\mathrm{m\,s^{-1}}$ intervals thereafter) and wind in the plane of cross section (arrows) for selected periods prior to and just after SEF. Arrow length in the vertical direction is scaled to be ten times longer than in the horizontal direction for the same flow speed.





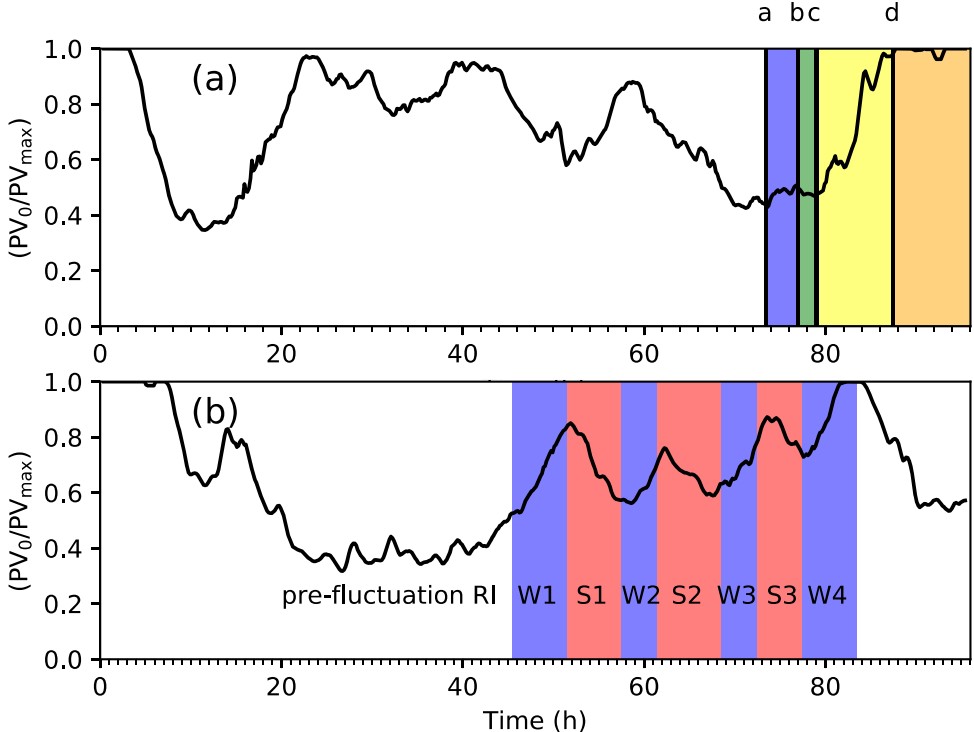

**Figure 8.** Ratio of the low–level PV (depth averaged between 1052 m and 4062 m) at the centre of the TC to the maximum azimuthally averaged low level PV for: (a) ensemble member 10 of the 05 September 12 UTC simulation showing the full eyewall replacement cycle, and (b) ensemble member 15 of the 03 September 00 UTC simulation showing the intensity fluctuations. In the case of (a) the vertical lines labelled a,b,c,d correspond to each of the four chosen times (rows) in Figure 1 for example vertical line c corresponds to Fig.1 e,f in 1. In panel (b) the weakening and strengthening phases are shown.

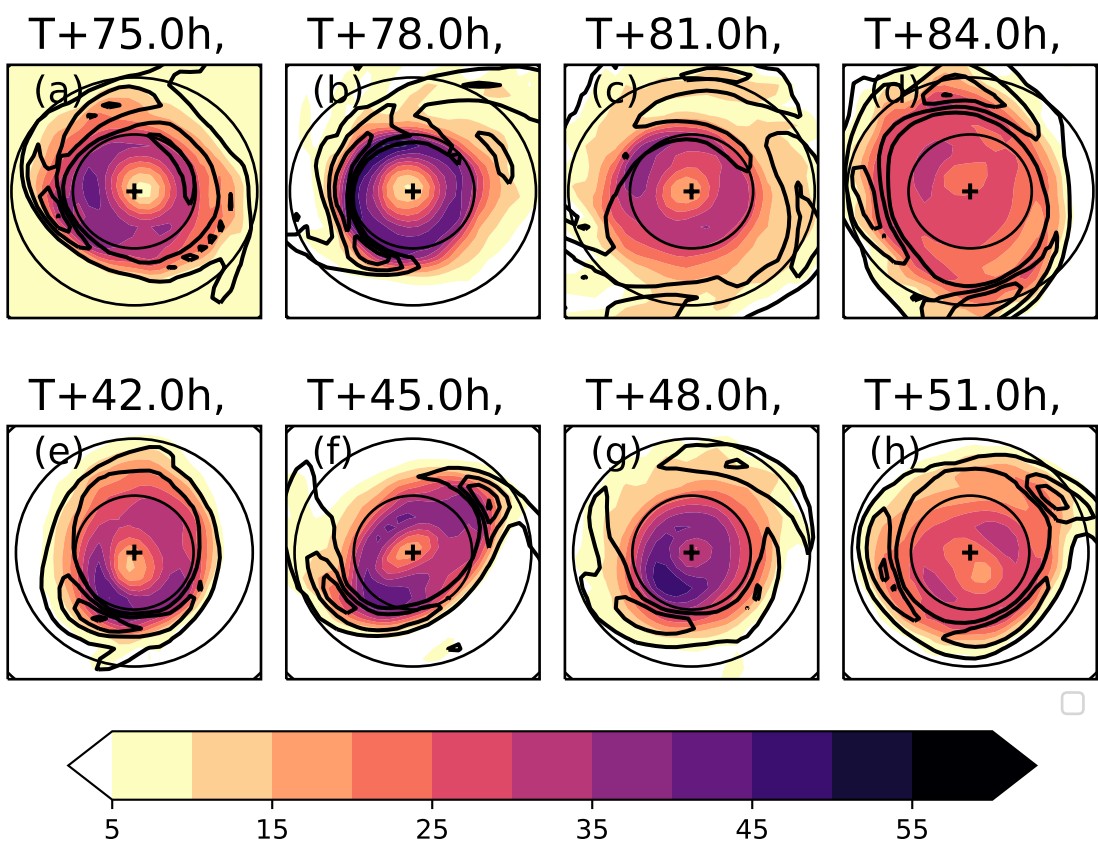

**Figure 9.** PV (PVU, shaded) at 1532 m height for selected times and vertical velocity ($1\,\mathrm{m\,s}^{-1}$, black contour). A black cross marks the centre of the storm. Black circles show radii in increments of 25 km. The top row is from ensemble member 10 of the 05 September 12 UTC simulation showing the full eyewall replacement cycle, the bottom row for ensemble member 15 of the 03 September 00 UTC simulation showing the intensity fluctuations.





**Figure 10.** Vertical velocity ($\mathrm{m\,s^{-1}}$, shaded) and radial wind (0.1, 0.2, 0.5, 1, 2, 5, 10, 20, 50 $\mathrm{m\,s^{-1}}$ positive (solid) and negative (dashed) black contours). Panels (a-d) from ensemble member 10 of the 05 September 12 UTC simulation showing the full eyewall replacement cycle and panels (e-h) for ensemble member 15 of the 03 September 00 UTC simulation showing the intensity fluctuations.



**Figure 11.** Agradient wind ($\mathrm{m\,s^{-1}}$, shaded), hourly tendency of the radial wind (0.5, 2, 5, 10 $\mathrm{m\,s^{-1}\,h^{-1}}$, positive (solid) and negative (dashed) yellow contours), tangential wind hourly tendency (0.5,2,5,10 positive and negative $\mathrm{m\,s^{-1}\,h^{-1}}$, black contours). Panels (a-d) from ensemble member 10 of the 05 September 12 UTC simulation showing the full eyewall replacement cycle and panels (e-h) for ensemble member 15 of the 03 September 00 UTC simulation showing the intensity fluctuations.






**Figure 12.** (a,c) Coloured contours show ventilation diagnostic index ($10^9 \, \mathrm{kg s^{-1}}$) with the azimuthally averaged radially integrated mass flux taken between a height of 6 and 1 km as a function of integration radius. Black contours show vertical velocity in $1 \, \mathrm{m s^{-1}}$ intervals and the $0.5 \, \mathrm{m s^{-1}}$ contour for 6 km height while the yellow contours show the same for 1 km height. (b,d) Coloured contours show the azimuthally averaged radial wind at 1532 m height (just above the boundary layer, $\mathrm{m s^{-1}}$) while black contours show the azimuthally averaged surface radial wind in $2 \, \mathrm{m s^{-1}}$ intervals (dashed contours indicate negative radial wind or inflow). (a,b) show these variables for the case of the eyewall replacement cycle while (c,d) show it for the case of the W1 intensity fluctuation.





**Figure 13.** Equivalent potential temperature (K, shaded), vertical velocity (0.1, 0.2, 0.5, 2, 5, 10 m s$^{-1}$ positive (solid) and negative (dashed) black contours). Panels (a-d) from ensemble member 10 of the 05 September 12 UTC simulation showing the full eyewall replacement cycle and panels (e-h) for ensemble member 15 of the 03 September 00 UTC simulation showing the intensity fluctuations. Also shown is the height dependent RMW (dashed grey line).





**Figure 14.** Schematic outlining the key differences between the modelled eyewall replacement cycle and intensity fluctuations of Hurricane Irma. The azimuthally averaged structure of the convection is indicated by the carton clouds, the secondary circulation indicated by the blue arrows, and tangential wind structure by dashed lines. The evolution is described at the start of (a) the weakening phase in the intensity fluctuations, and (b) the SEF in the case of the eyewall replacement cycle. Subsequent stages in the development are indicated for the eyewall replacement cycle (d,f), and the intensity fluctuations (c,e) with the approximate number of hours indicated by the time in the title.