# Peer review of "Comparing short term intensity fluctuations and an Eyewall replacement cycle in Hurricane Irma (2017) during a period of rapid intensification"

_EGUsphere, 2023_

## Referee Comment (RC2)

This paper examined SEF events simulated in a model ensemble and compared it to intensity fluctuation during RI. While the overall quality of the writing is commendable, there are some notable concerns related to the choice of comparison, missing components, and the comprehensiveness of the SEF theories under examination. Given the amount of effort needed to resolve these issues, I recommend rejection until the authors solve these fundamental issues.

Major issues:

1. **Motivation for comparing SEF with intensity fluctuation during RI**
     The first major issue of this study is the authors' choice and lack of clear motivation to compare the SEF which occurs *after RI*, with the intensity fluctuation that occurs *during the RI period* (i.e., section 3.3.1 to 3.3.4).
     These two types of events occur at different stages of the TC lifecycle (*during* versus *after* RI). The simulated SEF events occur near T+77h, with T here denoting the model initialization time at 05 Sept 1200UTC, which implies that the SEF roughly occurs near 08 Sept 1700UTC. On the other hand, the first episodes of simulated intensity fluctuation occur near T+44, with T here representing the model initialization at 03 Sept 0000UTC, which implies that the intensity fluctuation occurs near 04 Sept 200UTC. This means that these two events have a time difference of almost 3.5-4 days (about 93 hours). Although the authors did not show intensity time series in this manuscript, from their recently published Torgerson et al. 2023 and it is shown that the corresponding intensities of these two types of events are significantly different (i.e., 950 hPa for intensity fluctuation and near 910 hPa for SEF). This important information is *not* mentioned anywhere in this manuscript.
     Because of the large difference in time and intensity when the two types of events occur, many aspects of the storm are drastically different, such as inner-core wind structure, inertial and symmetric stabilities, relative humidity distribution, and very importantly the environment in which the storm was embedded (see my comment 4 below). It is only until very late in the manuscript that the authors showed the $\theta_e$ cross sections in Fig. 13, which gives the readers some senses of how different the TC vortex structure is between those two events. As in Fig. 13a-d, after RI (when SEF occurs), the $\theta_e$ at the TC inner core is significantly higher, with contours that are more vertically oriented. This $\theta_e$ structure indicates a more symmetrically neutral eyewall structure and must be associated with a significantly more intense tangential wind structure and inertial stability (which is not shown!), as well as less tendency for the inner eyewall to further intensify. In contrast, during RI, the TC intensity must still be below its potential maximum intensity and the primary eyewall has more conditional/symmetric instability. As shown in Fig. 13e-h, the eyewall $\theta_e$ during the early intensity fluctuation has a clear inward bending structure and decreases vertically near z = 2 - 5 km, indicating the eyewall has a greater conditional/symmetric instability and thus greater tendency to further intensify. However, the TC intensity and basic state vortex structure are not mentioned or examined (not in the discussion of Fig. 13, nor presented in any of the earlier figures), nor considered to be relevant factors contributing to the distinct behaviors in the intensity fluctuation.
     Fundamentally, given the large differences in the basic-state vortex structure and intensity, *why should we expect that the TC would undergo a similar evolution? In other words, what makes the authors think that these two types of events are comparable to one another?* To be honest, the entire section 3.3 reads like it is comparing apples with oranges and *then concludes that apples and oranges are different*. For example, near L254-268 where the authors discussed Fig. 9, they

compared the low-level PV evolutions between SEF and intensity fluctuation. The main findings (summarized below) are that

- The increase in azimuthal symmetry in the intensity fluctuation was attributed to the weakening of wave–2 inner rainband structures, but not present in the ERC.
- The changes in azimuthal PV symmetry during the ERC are smaller than the intensity fluctuations.
- The maximum standard deviation of PV does not change much during ERC, whereas for the intensity fluctuation, there is a rapid decrease in the standard deviation of PV during a weakening phase and an increase in the standard deviation of PV during a strengthening phase.

But what are the respective implications of these three findings? Given that in the SEF case, we know that there is an outer eyewall forming, which then undergoes ERC, whereas in the intensity fluctuation, there is no outer eyewall formation, all we can conclude here is that they evolve differently because they are fundamentally different processes that occur in different background vortex state. In fact, the authors also did not provide any further discussion about the implications of these differences.

Similar issues go with the subsequent sections. For instance, in L280-283 describing intensity fluctuation,

"*However, unlike in the case of the eyewall replacement cycle, there is no updraught (associated with outer rainbands) outside of the eyewall and above the boundary layer that moves radially inwards over time before merging with the boundary layer updraught associated with the newly forming secondary eyewall.*"

This is essentially saying that the intensity fluctuation is different from SEF because there is no mechanism to form a new eyewall. But, *do we need to compare SEF with intensity fluctuation in order to learn about the role of the developing rainband in the SEF process?* The answer is no because the importance of outer rainband development has been identified in many other studies (see my major comment 3), all of which did not compare SEF with intensity evolution to reach this conclusion. *So, what are we expecting to learn by comparing SEF with intensity fluctuation?*

Most of the discussion in sections 3.3.1 to 3.3.4 simply highlights the *intrinsic differences* between the SEF and intensity fluctuation, all of which are not surprising given they occur in substantially different intensities and vortex states and are undergoing distinct evolutions. The fundamental issues that need to be addressed here are *what are the motivations to do these comparisons*? *What are the justifications to convince the readers that these two types of events are comparable to one another, so that it is possible to identify a nontrivial cause leading to the observed differences?* And finally, *what meaningful conclusions can we draw based on the observed differences?*

**2. Confusing time referencing**

As I mentioned in major comment 1, it takes the reader quite some effort to compare different figures in the manuscript to realize the exact time difference between these events. Many of the Hovmoller diagrams start with some positive hours, such as 70 h in Fig. 2, 3, and 60 in Fig. 6. I presume for SEF-focused figures, these times are counted from the *initialization time at 05 Sept 1200UTC*. On the other hand, for intensity fluctuation-focused figures, such as Fig. 8b, 9e-h, 10a-h, and 11a-h, the times T+xh are counted from *the initialization time at 03 Sept 0000UTC*. This time labeling scheme is misleading, because, for example, it gives the impression that these two events are about 33 hours apart (comparing Figs 9a-d and 9e-h). However, because of the

initialization time difference, there is an additional 60 hours of time difference (i.e. time difference between 03 Sept 0000UTC and 05 Sept 1200UTC), making the total time difference of 93 hours time difference, which is almost 4 days. This approach of time referencing hides the large time difference between the two events, which is inappropriate. Therefore, the authors need to come up with a new time referencing approach that accurately describes the actual time difference between the two types of events. One possible approach is to use different labels in all figures that currently use the label $T$, such as $T_{0309\_00Z}$ and $T_{0509\_12Z}$ to represent the two initialization times.

**3. SEF mechanism related to outer rainband dynamics**

This study only examined two VRW-related hypotheses (i.e., VRW stagnation radius and filamentation) and boundary layer unbalanced processes. However, there is a large body of studies emphasizing the rainband-driven SEF mechanism. Here is listed just a small portion of them: Qiu and Tan 2013; Li et al. 2014; Zhu and Zhu 2014; Zhu et al. 2015; Tang et al. 2017; Chen 2018; Chen et al. 2018; Yu et al 2021a,b, 2022. The only study that the authors cited in the introduction is Didlake et al. 2018. But rainband-related SEF hypothesis was proposed as early as Didlake and Houze 2013 and has since been further developed by many subsequent modeling and observational studies.

For instance, Qiu and Tan (2013) examined the connection between unbalanced boundary layer response to asymmetric inflow induced by the outer rainband. This asymmetric inflow is also similarly identified in observation (Didlake and Houze 2013; Didlake et al. 2018) and many modeling studies (Dai et al. 2017; Zhang et al. 2017; Chen 2018; Chen et al. 2018; Yu et al. 2021a,b and 2022), and is referred to as "mesoscale descending inflow" (Didlake and Houze 2013; Didlake et al. 2018; Yu et al. 2021a and 2022). Yu et al. 2021a and 2022 demonstrated the important role of a mesoscale descending inflow within the stratiform precipitation region in initiating the broad-scale wind field acceleration and boundary layer cold pool dynamics in sustaining convective updraft at the inner edge of the descending inflow. From observation, it has even been demonstrated that 79% of observed SEF events have a stationary rainband complex within 6 hours of the SEF development (Vaughan et al. 2020). Given all these important rainband-focused SEF research and the apparent importance of rainband processes in the present case, I am surprised that the author neglected this large body of literature and only focused on examining the few hypotheses that do not emphasize rainband dynamics.

Here, I want to clarify that I do agree that the unbalanced boundary layer process is an important part of the SEF mechanism. However, it should be noted that the unbalanced boundary layer process argument does not emphasize the importance of the rainband process *as a precursor of SEF*. Given that in the present case, the rainband development precedes the onset of SEF (or as the precursor of SEF), it is unreasonable not to examine whether the simulated SEF event shares similarities with the hypotheses and findings from the other rainband-related SEF studies. Therefore, the entire section 3.2 seems to be incomplete to me.

**4. Environmental wind shear is the largest organizing factor of the TC rainband development**

Given the importance of rainband development in the simulated SEF case, it is important to show the environmental controlling factors related to rainband development. Specifically, it is well known that the largest organizing factor of rainbands is environmental shear. However, this study mentions nothing at all about the role of environmental shear. How does the environmental shear vary over time, particularly during the two periods of intensity fluctuation events and SEF? Are

there any differences in shear magnitude and direction in shear during these two types of events? Based on a quick search of relevant publications, Fischer et al. 2020 showed that the 850-200 shear increases to about 5 $ms^{-1}$ between 07-08 Sept during the SEF event, while near 04-05 Sept the VWS was only about 2-3 $ms^{-1}$ (see their Fig. 3a). This increase in VWS is likely an important environmental controlling factor that contributes to the development of outer rainband in the SEF event (both actual or simulated). This also explains why the rainband during RI is much less developed due to weak shear in the environment. Again, this important piece is entirely missing in this study.

**5. Emergence mechanism of supergradient wind**

While I agree that the unbalanced boundary layer process is an important part of the SEF mechanism, one major issue of the argument is that the boundary layer spin-up mechanism does not explain what exactly drives the emergence of supergradient wind. As stated in L388, the authors do not seem to delve into the detailed analysis to determine what causes the enhanced inflow, but merely point to the positive feedback between rainband and boundary layer processes, i.e., expanded tangential wind field, enhanced boundary layer inflow and the subsequent emergence of supergradient wind and updraft? I would say this feedback mechanism is not new. If the authors are reluctant to determine the cause of enhanced boundary layer inflow, then what is exactly new here?

**6. Other causes of intensity fluctuation**

For pre-intensifying TC in shear, intensity fluctuation is often caused by the inward intrusion of low-$\theta_e$ air into TC's inner core, a process known as ventilation. This ventilation is not the same as the mass ventilation out of the boundary layer (diagnosed in section 3.3.3) that relates to the deepening of surface pressure. Low-$\theta_e$ ventilation is a well-known factor that can cause the weakening of TC (Tang and Emanuel, 2010, 2012a,b) or lead to delay of intensification, specifically for TCs in shear that are before reaching its mature state. Specifically, it is known that there exist several pathways of ventilation, namely the mid-level radial ventilation and the low-level downdraft ventilation. Alland et al. 2020a,b showed a nice summary of these pathways. A recently published study (Yu et al. 2023) also show how low-$\theta_e$ ventilation into TC inner core can cause substantial weakening of TC intensity. These possible mechanisms of intensity weakening are not examined or mentioned in the manuscript.

Overall suggestions to resolve these major issues and improve the manuscript:

Based on the major issue discussed above, I strongly suggest the authors rethink the necessity of comparing SEF events with intensity fluctuation that occurs in a much weaker intensity state. At this point, this manuscript is structured in a way that the comparison is not necessary. If this paper aims to focus on examining the SEF process, I strongly suggest the author look into the large body of rainband-related SEF studies and delve into the rainband structure to examine the linkage and feedback between TC rainband and unbalanced boundary layer processes. If this paper aims to focus on intensity fluctuation, then I also agree with Dr. John Methven's comments about the need to distinguish the current paper from the authors' recently published 2023 paper.

Minor comments:

Title: The title is incorrect. The eyewall replacement cycle does not happen *during* a period of rapid intensification period, but *after* the rapid intensification.

L11: Nascent TC is susceptible to environmental shear, which could cause a substantial delay in intensification. During this delay, intensity can remain nearly steady as the TC vortex undergoes precession. So the statement that the intensity of newly formed TC typically increases over time is incorrect.

L38: outflow . Jet usually is used to describe localized enhanced wind fields. For TC vortex, the strongest wind is mostly likely along the tangential direction, e.g., supergradient jet.

L41-46: This paragraph aims to provide the motivation, but is weak. Only stating that this study aims to compare these two events is not sufficient. As explained by my major comment 1, the background state TC vortex is drastically different, making a direct comparison of these two events nearly impossible to extract meaningful conclusions, other than saying that they are intrinsically different processes that evolve entirely differently. This paper needs to substantially strengthen the motivations by providing stronger reasoning, such as explaining why the authors think that these two processes are comparable. Why is it necessary to compare these two processes?

Introduction: Overall, the discussion about rainband dynamics in the introduction is not enough. There is only one sentence in L25-27 citing Didlake et al 2018. Given that rainband dynamics plays a clear role in the simulated SEF event, the authors need to provide a brief survey about rainband dynamics in a sheared environment to allow readers background information about the role of rainbands in the SEF process, a summary of what previous rainband-related SEF studies had learned (see major comment 3), and what still remains unclear.

L60: that's brightness is inversely correlated to → which has brightness inversely correlated to

L99: The SEF event happened earlier in reality → The *observed* SEF happened earlier in reality.

Section 3.1: If this study aims to compare the SEF with intensity fluctuation, this section should not only focus on the SEF event but also provide basic information about the vortex structure during intensity fluctuation. Specifically, a comprehensive comparison of the vortex structure (e.g., tangential wind, moisture) when these two events happen is also necessary to show readers the basic vortex structures when the two events happen and how comparable they are. Also, discussions about the difference in intensity (showing intensity time series) and environmental wind shear (also time series) are necessary.

Section 3.2: This list of SEF mechanisms is not comprehensive. SEF-related mechanisms also need to be examined. See major comment 3.

L259-260: A theory of barotropic instability across the moat in the ERC process (Lai et al. 2019; Lai et al 2021a,b) has been proposed in recent years that explain the elliptic shape of the PV

structure during ERC, as well as the rapid decay of the inner eyewall, which are clearly relevant to this study.

Figures: Many of the plan-view figures do not have spatial markers in the x and y axes. Even though the authors added circles representing 25 and 50 km radii, but showing markers in the axes can let the readers see the exact size more clearly. Also, the black circles overlap with the black contours, making them difficult to see (e.g, Fig. 9). Also, in Fig. 1, there is no circle nor markers in the axes, which are absolutely needed. How can we tell if the simulated TC size is realistic compare to the observation?

**References mentioned:**

Alland, J. J., B. H. Tang, K. L. Corbosiero, and G. H. Bryan, 2021a: Combined effects of midlevel dry air and vertical wind shear on tropical cyclone development. Part I: Downdraft ventilation. J. Atmos. Sci., 78, 763–782, https://doi.org/10.1175/JAS-D-20-0054.1.

Alland, J. J., B. H. Tang, K. L. Corbosiero, and G. H. Bryan, 2021b: Combined effects of midlevel dry air and vertical wind shear on tropical cyclone development. Part II: Radial ventilation. J. Atmos. Sci., 78, 783–796,https://doi.org/10.1175/JAS-D-20-0055.1.

Chen, G., 2018: Secondary eyewall formation and concentric eyewall replacement in association with increased low-level innercore diabatic cooling. J. Atmos. Sci., 75, 2659–2685, https://doi.org/10.1175/JAS-D-17-0207.1.

Chen, G., C. C. Wu, and Y. H. Huang, 2018: The role of near-core convective and stratiform heating/cooling in tropical cyclone structure and intensity. J. Atmos. Sci., 75, 297–326, https://doi.org/10.1175/JAS-D-17-0122.1.

Dai, Y., S. J. Majumdar, and D. S. Nolan, 2017: Secondary eyewall formation in tropical cyclones by outflow–jet interaction. J. Atmos. Sci., 74, 1941–1958, https://doi.org/10.1175/JAS-D-16-0322.1.

Didlake, A. C., and R. A. Houze Jr., 2013: Dynamics of the stratiform sector of a tropical cyclone rainband. J. Atmos. Sci., 70, 1891–1911, https://doi.org/10.1175/JAS-D-12-0245.1.

Fischer, M. S., R. F. Rogers, and P. D. Reasor, 2020: The rapid intensification and eyewall replacement cycles of Hurricane Irma (2017). Mon. Wea. Rev., 148, 981–1004, https://doi.org/10.1175/MWR-D-19-0185.1.

Li, Q., Y. Wang, and Y. Duan, 2014: Effects of diabatic heating and cooling in the rapid filamentation zone on structure and intensity of a simulated tropical cyclone. J. Atmos. Sci., 71, 3144–3163, https://doi.org/10.1175/JAS-D-13-0312.1.

Qiu, X., and Z. M. Tan, 2013: The roles of asymmetric inflow forcing induced by outer rainbands in tropical cyclone secondary eyewall formation. J. Atmos. Sci., 70, 953–974, https://doi.org/10.1175/JAS-D-12-084.1.

Tang,X., Z. Tan, J. Fang,Y.Q. Sun, and F.Zhang, 2017: Impacts of the diurnal radiation cycle on secondary eyewall formation. J. Atmos. Sci., 74, 3079–3098, https://doi.org/10.1175/JAS-D-17-0020.1.

Tang, B., and K. Emanuel, 2010: Midlevel ventilation's constraint on tropical cyclone intensity. J. Atmos. Sci., 67, 1817–1830, https://doi.org/10.1175/2010JAS3318.1.

Tang, B., and K. Emanuel, 2012a: Sensitivity of tropical cyclone intensity to ventilation in an axisymmetric model. J. Atmos. Sci., 69, 2394–2413, https://doi.org/10.1175/JAS-D-11-0232.1.

Tang, B., and K. Emanuel, 2012b: A ventilation index for tropical cyclones. Bull. Amer. Meteor. Soc., 93, 1901–1912, https://doi.org/10.1175/BAMS-D-11-00165.1.

Vaughan, A., Walsh, K. J. E., &Kepert, D. J. (2020), The stationarybanding complex and secondaryeyewall formation in tropical cyclones. Journal of Geophysical Research:Atmospheres,125, e2019JD031515.https://doi.org/10.1029/2019JD031515

Yu, C.-L., A. C., Didlake Jr., and F. Zhang, 2021a: Asymmetric rainband processes leading to secondary eyewall formation in a model simulation of Hurricane Matthew (2016). J. Atmos. Sci. 78, 29-49.

Yu, C.-L., A. C., Didlake Jr., F. Zhang, and J. D., Kepert, 2021b: Investigating axisymmetric and asymmetric signals of secondary eyewall formation using observations-based modeling of the tropical cyclone boundary layer. Journal of Geophysical Research: Atmospheres. 126. 10.1029/2020JD034027.

Yu, C.-L., A. C., Didlake Jr., and F. Zhang, 2022: Updraft Maintenance and Axisymmetrization during Secondary Eyewall Formation in a Model Simulation of Hurricane Matthew (2016). J. Atmos. Sci. 79, 1105-1125.

Zhu, Z., and P. Zhu, 2014: The role of outer rainband convection in governing the eyewall replacement cycle in numerical simulations of tropical cyclones. J. Geophys. Res. Atmos., 119, 8049–8072, https://doi.org/10.1002/2014JD021899.

Zhu, P., and Coauthors, 2015: Impact of subgrid-scale processes on eyewall replacement cycle of tropical cyclones in HWRF system. Geophys. Res. Lett., 42, 10 027–10 036, https://doi.org/10.1002/2015GL066436.

---

## Author Comment (AC1)

**Reply to the reviewers' comments for manuscript egusphere-2023-1272 submitted to Weather and Climate Dynamics**

We would like to thank the two anonymous reviewers and John Methven for their insightful and useful comments. We have revised the manuscript extensively in response to these comments. We think that the manuscript is much clearer as a result. In particular, all three reviewers raised concerns about the motivation for our comparison of an eyewall replacement cycle with an intensity fluctuation and a need to highlight the novelty of this study. In our response we have been careful to highlight our hypothesis that while there are obvious differences between the two phenomena in terms of when they occur in the TC lifecycle, both lead to intensity changes in the storm as a result of rainband activity and therefore it is a legitimate scientific question to ask whether there are similarities in the underlying mechanisms even if there are differences in the outcomes due to the different location of the rainbands and differences in the storm due structure to the stage of the lifecycle the storm is in. The reviewers also highlight several points where they felt information was missing in the explanation. In quite a few places this information is in our related previous paper (Torgerson et al, 2023) which uses the same model setup and simulations. We had tried to avoid too much duplication of information but acknowledge that we might have been too zealous in this and have added back additional information at a number of places to aid the reader without them needing to refer back to our previous paper.

There are also questions pertaining to the novelty of our results which we consider to be an omission of clarity on our part. To be clear, the novelty of our results lies in identifying two superficially dissimilar types of intensity fluctuation and propose they are connected by a common initiating mechanism involving a dynamic adjustment to convection outside the eyewall which leads to a runaway positive feedback process within the boundary layer. We have also proposed that the differences can largely be explained by the large radial separation and the existence of a moat region in the case with an eyewall replacement cycle that does not exist in the case of the short-term intensity fluctuations.

We provide a point-by-point response to the reviewers' comments below. The reviewers' comments are in black, and our responses are in blue.

**Reviewer 1**

The story is not well told. The background of Irma, whether it's influenced by the environment, how the TC intensity evolves in both observation and simulations, are all missing.

This background information on Irma is contained comprehensively in our previous paper (Torgerson et al., 2023) which focuses on the intensity fluctuations themselves and is based on the same model data. We wanted to avoid too much repetition between the two manuscripts; however we do accept that a basic summary is required here. Clarification on the storm evolution and its synoptic environment has been added in the model evaluation subsection.

The observation is completely not necessary. It is not used to validate the simulation, and the forecast result is different from the observation anyway.

The purpose of the observational data is to justify the use of the model simulations as a means of understanding the mechanisms behind the fluctuations in relation to the eyewall replacement cycles. As such it is important that the model simulations are adequately capturing both types of fluctuations such that any conclusion made on the basis of the simulations can reasonably be expected to also apply to a real storm. Although the observations and the model simulations are never going to agree completely; key features were captured which gives us confidence in the results. Further clarification has been added to the model evaluation subsection.

Justification of picking and comparing the two members is missing: are they similar in structure before SEF? It seems not according to Figure 13, the theta_e difference is huge. What's the purpose of comparing these two ensemble members? What are the behaviours of other ensemble members? Why are the results in this case study unique from other case studies?

In a real storm it is difficult to compare the two types of fluctuations for the same storm structure. Our hypothesis is that the short-term intensity fluctuations occur *during* RI while an eyewall replacement cycle occurs at the end of RI (or indeed causes the end of RI) which means for the same storm the fluctuations are going to occur at different stages of development and hence the storm intensity, theta_e and other general characteristics will differ. The two ensemble members presented were chosen because they show clear examples of an eye wall replacement cycle and a short-term intensity fluctuation at the correct stage in the storm's development and these are qualitatively similar to the observations. Other ensembles also show one, both or neither type of fluctuation occurring. Since both types of fluctuation are sensitive to stochastic processes this variability between ensemble members is to be expected. The unique part of the study is linking two seemingly disparate phenomena to the same initial dynamical response to convection outside of the eyewall. We have added a paragraph to the methodology section explaining this choice. We are preparing a future paper that will extend this work to other storms with the aim of providing evidence that the concepts developed are transferable to similar storms in similar stages of development.

**Minor comments**

-Figure 1, the time over each panel is very confusing, they are not the same as the other figures.

This is explained in the figure caption and surrounding text. The time given here is relative to the SEF stage of development rather than with respect to model initialization. The purpose of this is to allow a more direct comparison between observational and model data. The rest of the paper largely focuses on analysing the model simulations and so we use the time since model initialisation in the other figures. We appreciate that this might be confusing to a reader and so we have revised the figures and captions to be more explicit about which simulation / which times are being shown in the plots. The notation we have used in the revised paper and this document is as follows:

$T_{SEF}$ is the reference time with respect to the qualitatively observed SEF from the microwave data. T+0.0h in Fig.1b,d,f,h of the original draft is redefined as $T_{SEF}$ occurring on 07 September 2017 1015UTC .

$T_1$ is the reference time with respect to the first run initialized on 03 September 2017 00UTC (ensemble member 15). For example $T_1$+24.0h is defined to be 04 September 2017 00UTC. Fig.9 (e-h) of the original draft use the $T_1$ reference.

$T_2$ is the reference time with respect to the second run initialized on 05 September 2017 12UTC (ensemble member 10). For example 04 September 2017 00UTC is defined as $T_2$-36 hours with respect to this reference. The majority of figures in the original draft were defined with respect to $T_2$ including Fig. 2.

-Inaccurate or confusing expressions:

In the title: eye wall replacement cycle cannot happen 'during' a period of rapid intensification.

The title has been changed to "Comparing short term intensity fluctuations and an eyewall

replacement cycle in Hurricane Irma (2017) during *and immediately after* a period of rapid intensification" to emphasise that the EWRC occurs after the period of RI.

L11: 'typically increase over time'?

Changed to improve wording. 'There is a tendency overall for mean sea level pressure to fall and tangential wind speed to increase'.

L11-L15: the description here only fits the situation with no hostile environmental conditions.

Changed to make this clear. 'Even in the case of an approximately symmetric tropical cyclone in an ideal environment (i.e. minimal wind shear)...'

L37-L38: not very sure what this sentence means.

Changed to make this clearer. 'The inner rainband convection was proposed to cause an acceleration of outflow above the boundary layer, through dynamical adjustment, leading to a weakening of the tangential wind within the eyewall.

**Comments from John Methven**

In this new paper the authors need to make much clearer in their introduction what the novel aspects are in this study and how it relates to their last paper. This is not clear from lines 35-46 on p.2.

The central novel aspect of this study is that two disparate types of fluctuations, i.e. SEF and short-term intensity fluctuations during RI, are linked by a similar initiating

mechanism which involves a dynamical response to rainband convection followed by a runaway positive feedback within the boundary layer. We made changes to the manuscript to emphasise that the motivation for the comparison between the two forms of intensity fluctuations is the similarity between the dynamical processes that occur in the inner or outer rainbands of the fluctuations and ERCs respectively. In addition, we also emphasised the novel results in the paper.

Although I don't dispute this distinction it did seem a bit limited to me. The big question is then why an outer convective rainband is forming in one case and not in the other.

We have no reason to believe that the formation of an outer rainband is not possible in the case of the short-term intensity fluctuations or even that the two types of fluctuations are mutually exclusive. Indeed, there are ensemble forecast that seem to show both fluctuations happening at the same time which is possible for a TC that has outer and inner rainbands. We also note that, given the ERC is a more dramatic structural realignment and that there is greater separation between the eyewall and the SEF region (compared to the inner rainband region), the development of a strong enough outer rainband to trigger the series of processes starting the SEF takes considerably longer. The 'irreversible' structural change in an ERC is also relevant as it allows intensity fluctuations to potentially occur multiple times during RI while an ERC would tend to terminate RI. We have clarified throughout the paper that the study is focused on the similar processes that occur in the outer and inner rainbands and have also added a paragraph in the discussion noting that the two types of fluctuations are not mutually exclusive.

I was not convinced by the authors' arguments that the VRW outwards propagation was not a reason for the distinction between the two types of events. Figure 2 seemed to indicate VRW events preceding vertical velocity increase at the outer radius. I could not understand why figure 3 was truncated early without extending to T+85h, missing the vital period 80-85 hours when more VRWs reached the stagnation radius.

We agree that a VRW at T+79h (Fig. R1) did accelerate the tangential wind near the stagnation radius, which in turn may have played a contributing factor by enhancing frictionally induced convergence in the outer core. However, this acceleration of the tangential wind speed, consistent with the VRW was small compared to the original VRW described in the study that occurred at T+73h, between T+74h and T+77h no tangential wind acceleration occurred near the stagnation radius. The SEF occurred between T+77h and T+78h. So, while VRW activity may have occurred after T+80h it cannot have played a role in the instigation of the SEF. We have added that VRWs do play a role in the SEF event and how it synergises with other proposed mechanisms.

In particular, Section 3.3 contains a lot of discussion and description of different diagnostics shown but I did not think the discussion was rigorous enough to be very compelling in terms of distinguishing mechanisms at play. I would recommend trying to reduce the length of this sub-section and diversity of diagnostics shown and homing in on the key results.

We have introduced the subsections in 3.3 in a way that much more effectively introduces each diagnostic and explains how it helps us understand the similarities or differences in mechanisms between the EWRC and short-term intensity fluctuations. Specifically, the utility of the metrics in section 3.3 are described below in the order they appear in the study.

- $PV_0/PV_{max}$ determines the radial PV distribution. Higher values of this metric indicate more of a monopole like structure while lower values correspond to move of a ring like structure. One process that occurred in the short-term intensity fluctuations was a collapse in the eyewall structure proposed to be linked to a barotropically unstable state. In the SEF formation part of the ERC this did not occur showing that this process was delayed and not linked to the cause of the SEF.
- Mass ventilation index provides a means to understand why the outflow jet above the boundary layer changes during the two types of fluctuations and the implications of this change. Namely we saw a similar tendency in both types of fluctuations with regions developing in both cases where the convection became increasingly unable to ventilate the near surface mass influx adding weight to the common mechanism between the fluctuations.
- The equivalent potential temperature was used to compare the thermodynamic structures in the two different fluctuations with the aim of understanding why there was no inner eyewall decline in the short-term fluctuations. The salient point was a lack of a cold dry 'reservoir' from the moat region.

We have also added a paragraph at the start of section 3.3 that explicitly references the key mechanism in terms of the unbalanced feedback response that occurs in both types of intensity fluctuation.

Section 2.2: This section lacked detail about the nature of the ensemble perturbations in the initial conditions and also the use of lateral BCs in the LAM. Presumably the ICs are interpolated from the parent global model onto the LAM grid and are therefore relatively smooth? Are any perturbations applied on smaller scales? Is there is a stochastic parametrization scheme operating?

The model setup was identical to that used in Torgerson et al. (2023), since this is a continuation to the work done in that paper; these details are available in section 3.3 of that paper, however we now also include more details in the current manuscript. The lateral boundary data is supplied from the global model to the regional model at a three-hour frequency (Short et al. 2018) while the initial conditions are interpolated from this same global model. Stochastic perturbations are applied within the boundary layer parametrization (Bush et al. 2020); there are also perturbations within the other parametrization schemes used to trigger the global model ensemble conditions (Sanchez et al., 2016). We did incorrectly refer to section 3.5 rather than section 3.3 of Torgerson et al. (2023) when the relevant information is given. This is now corrected.

Section 3.1: "Ensemble member that best matched observations". Seems like a valid approach but in what measures was it best?

We have emphasised that Fig. 1 represents the criteria used to make this decision. This ensemble member not only produced the most similar time scales of the four distinct stages given in the figure, but the convective pattern was also most similar with individual features like scattered cellular convection, spiral banding or ring banding concordant between the right and left panel (albeit sometimes rotated) and with the overall size scales of the convection (e.g. the size of the outer eyewall ring) also extremely similar. We have significantly strengthened our justification of the choice in the manuscript to reflect this.

l.130: "eyewall replacement cycle of Irma is captured well". How can you say that? On what basis?

This comment was made on the basis of the same criteria given in the previous response. We have expanded the final paragraph at L 130 to make this clearer.

l.154-155: This seems to be crucial to the whole paper. You state that the azimuthal phase velocity of the VRW was consistent with the dispersion relation. This is really important to demonstrate that the theory is relevant and that the disturbances in the model can be described as vortex Rossby waves. Please show the evidence. Similarly, you indicate the radial propagation of a VRW packet on Fig. 2 for one event. How did you calculate the curve? Why not do the same for the later events at T+75, 77, 79, 81, 83? They all seem to be similar in terms of PV disturbance and its radial propagation rate, even though in the text you write that "they do not propagate in a way consistent with the dispersion relation". It looks to me that they are very similar. Also, I would make the dashed line yellow or some other colour that would stand out on red-blue shading.

More details about the use of the dispersion relation to diagnose vortex Rossby waves, as used in this paper, is available in the PhD thesis Torgerson (2021) *Intensity fluctuations in Hurricane Irma (2017)* which is available online (https://etheses.whiterose.ac.uk/29953/). Section 3.5 of the PhD thesis describes the method first using the dispersion relation for vortex Rossby waves first derived in Montgomery and Kallenbach (1997). Equation 3.4 shows a dispersion relation for the angular frequency of vortex Rossby waves which can then be converted to an azimuthal phase velocity or radial phase velocity (equation 3.5) or height phase velocity (although this tends to be very noisy and not that usable) and compared to the phase velocities within the model. An agreement between the dispersion relation and the phase velocity of the PV anomaly in the model output is taken to be good evidence that the anomaly is caused by a vortex Rossby wave. *We* have added two additional candidate VRWs on the radial dispersion relation validation and also included a plot for azimuthal validation (Fig. R1 and R2*).* Note that for radial validation PV is plotted at a fixed (eastern) azimuth while for *azimuthal validation the radius is fixed at 75km (where VRW is high). The* start times for the radial and azimuthal dispersion relation *timelines* are slightly different since it takes time for the VRW to travel to 75km radius. We have added a summary of this information in section 3.2.1 and referenced the thesis where the technical details of the calculations are available. *Figure R1 appears in the manuscript but we have not*

*included figure R2 in the manuscript as it provides no useful or interesting information beyond providing evidence that the PV anomalies move in a way that is consistent with VRW dispersion and we consider the technical details offered in the reference sufficient.*

[Figure]

Figure R1: Wavenumber–2 PV for the eastern azimuth (shaded, PVU) at 1532 m as a function of time. Vertical velocity (black line contours of 0.4, 0.8, 1.6, 3.2 ms−1) along the same azimuthal angle and height. Also shown are the RMW for the same height (grey solid line) and 3×RMW as a proxy for the stagnation radius (purple solid line). The yellow dashed line shows the trajectory of a hypothetical VRW propagating from the RMW at T+71 h using the VRW dispersion relation.

Figure R2: Wavenumber–2 PV for the 75km radius (shaded, PVU) at 1532 m as a function of time. The black dashed line shows the trajectory of a hypothetical VRW propagating from the RMW at T+71 h using the VRW dispersion relation.

Of the three VRW candidates the originally depicted VRW event seems to match up best with the azimuthal and radial dispersion relations however the other two VRW candidates also show reasonable agreement, hence we agree that these additional two events prior to the SEF are plausible VRW candidates and as such we do admit that the VRWs played a bigger role than we originally suggested in broadening and accelerating the wind field prior to the SEF. In the case of the third VRW candidate just prior to the SEF there is a reasonable possibility that the acceleration of the wind field inside of and near the stagnation radius between T+77 and T+78h is caused by VRW activity although the first VRW candidate (although weaker) does generate a bigger tangential wind acceleration (see Fig.3 from the paper). Some further discussion has been added on

these points in the second paragraph of section 3.2.1 with attention given to the other two VRW candidates

Figure 3: This really needs to extend to the same point in the forecast as Fig.2. It seems to be missing all the action when the stagnation radius jumps outwards. It looks to me that there is tangential wind deceleration coinciding with the approach of VRWs towards the stagnation radius. It is hard to understand why it looks similar to me, but you state that it is not similar. Also, it seems that there is periodicity in the tangential wind acceleration events near the stagnation radius that is similar to the interval between the VRW events even if the phasing seems variable. I am sure some other readers will be picking up on the same things as me.

We have extended figure 3 to include the hours after SEF as in Fig. 2. The original VRW event is shown in Fig. 3 to cause a tangential wind acceleration of around 1.5-3ms$^{-2}$ near the stagnation radius for an hour between T+73h and T+74h. The weak deceleration near T+75h was only very briefly as high as 1ms$^{-2}$ near 100km radius and near the stagnation radius it is even less than this. It is difficult to associate this very small deceleration with any VRW event that is consistent with the dispersion relation. The notable aspect is a large acceleration that is associated with a VRW event but that occurs a few hours prior to SEF. We have emphasised this magnitude of acceleration in the text.

Figures 4 and 5. It would be better if the panels were at the same times as the other terms in the tangential wind budget?

The reasoning for the different choice of times is that the key points in the BSA mechanism and the VRW development do not occur at the same time. Different times are picked to best highlight the relevant processes.

l.166-167: I don't follow how you deduce that the VRW is not the direct cause of the SEF.

Figure 4 shows that there is evidence the described VRW event leads to a significant anomaly of positive tangential acceleration near the stagnation radius from eddy advection of absolute angular momentum at the time corresponding to the incipient VRW arriving at the stagnation radius. This does not occur, however, immediately prior to the SEF event itself suggesting that the proposed mean-eddy interaction that led to the acceleration hours prior to the SEF did not also cause the acceleration directly associated with the SEF itself. Further explanation has been added and linked back to the original Montgomery and Kallenbach (1997) paper.

l.183: "inertial parameter" -> "absolute vorticity of the azimuthally averaged state"

Changed.

l.196: What do you mean by "broadening of the wind field"? You need to be more precise here.

This has been more precisely defined at its first mention at the start of the SEF dynamics section in a footnote. We define broadening as 'a preferential increase in the outer core tangential wind field such that the negative radial tangential wind gradient is reduced.'

l.199: Define agradient wind and supergradient wind, perhaps with aid of some equations.

As above we have included a footnote explicitly defining both agradient and supergradient wind mathematically. 'A supergradient wind is a positive agradient wind, where the agradient wind is defined as $v_{ag} = v$ - $v_{gr}$ where $v_{gr}$ is the gradient wind satisfying the gradient wind balance $\frac{1}{\rho}\frac{\partial p}{\partial r} = \frac{v_{gr}}{r} + fr$ where ρ is the density of air, $p$ the pressure and $r$ the radial distance from the centre.'

Section 3.2.3: I don't follow the mechanism outlined at the end of this section. How can "increased agradient outward forces" "promote convection in the SEF region"? Why does "broadening of the tangential wind occurring above the boundary layer" result from "balanced dynamical adjustment to outer-rainband activity"? You would need to explain more clearly how these mechanisms work to convince the reader that they describe the evolution observed. As far as I am concerned there is not enough evidence to rule out the VRW mechanism as the reason for initiating secondary eyewall formation.

Increased agradient wind leads to greater frictionally induced convection within the boundary layer (reduction in radial inflow gradient) which through continuity leads to ascent, and in turn moist convection. One impact of the rainband is a modification of the secondary circulation above the boundary layer leading to increased AAM advection towards the SEF region. In the revised manuscript much more detail has been added clarifying these points. VRWs are also given greater relative importance within the entire process as a contributing factor to the wind broadening.

Section 3.3.1: Missing some rationale here. Why look at PV structure and what is the hypothesised behaviour in terms of PV in the case of a PV ring or PV monopole? For example, one aspect is that a PV ring can be unstable with respect to waves around the ring (e.g., Schubert et al, 1999, JAS) while a monopole can support waves but is stable.

Your example is prescient and is something we have referred to in our previous paper on short-term intensity fluctuations. A paragraph has been added at the start of this

section motivating the inclusion of PV distribution diagnostics. 'In Torgerson et al. (2023) one aspect that proved important in causing the fluctuations was an implied increase in barotropic instability during strengthening phases that culminated in a breakdown of the annulus like PV structure causing it to become more similar to a monopole. In order to assess whether the same process occurs prior to the SEF, the evolution of the radial PV structures are examined.'

Section 3.3.2 and Figure 11. How do all the things mentioned relate? It feels like equations are needed to make this more comprehensible.

To clarify these concepts we have more comprehensively introduced the agradient wind at the start of section 3.2 including a footnote that explicitly includes an equation for the agradient wind and should aid the reader in interpretating Fig. 11 when the concept is again used in section 3.3.2. Additionally the start of section 3.3 now includes a paragraph that contextualizes the use of agradient wind in terms of the proposed common mechanism between the two types of fluctuations.

Section 3.3.3: If mass ventilation was shown in Torgerson et al. (2023) why show it here? I was lost as to the purpose of figures 11, 12 and 13 and the related discussion. What does it all tell us? Basically I could follow Section 3.2, but I found it very hard to follow Section 3.3 and what the diagnostics tell us. The diagnostics were described but the interpretation and discussion was not rigorous enough to be very convincing. If you have equations in mind when you write the interpretation, then put them in the paper and sharpen the arguments rather than trying to say everything in words alone.

In Torgerson et al. (2023) mass ventilation was used as a diagnostic to understand the weaking process that occurs during the short-term fluctuations. The purpose of using that diagnostic here is to generalize this concept and extend it to SEF to identify differences and similarities. A key result is that a similar and stronger weakening in the ERC example is caused by the moat region which is unable to ventilate the incoming mass flux. The moat is a region of very weak ascent or descent that necessarily forms during the ERC process but does not form in the case of a short-term fluctuation; as such the inability to ventilate and the resultant impact on the outflow is stronger in the case of the SEF. Fig. 11 likewise focuses on the unbalanced boundary layer processes as a whole during the two different types of fluctuations. The key result highlighted here is that during the SEF there are much larger regions of supergradient winds outside the primary eyewall compared to the short-term fluctuation; yet the increased outward influence of the supergradient winds is a similarity shared between both fluctuations. Finally, Fig. 13 looks at the demise of the inner eyewall which is one process to which there is no clear analogue within the short-term intensity fluctuations. Clarification on this point has been added including a motivating paragraph at the start of section 3.33 that justifies the use of the metric and explains why it is important for the comparison in terms of the similar initiating mechanisms and direct consequences of that in terms of

the unbalanced boundary layer processes and its relationship with the free vortex above.

Appendix A: This is fairly standard and it would be sufficient to refer to the associated papers with the definitions.

We have removed this Appendix and instead given the relevant references, including to the PhD thesis which has the full method.

**Reviewer 2**

**1. Motivation for comparing SEF with intensity fluctuation during RI** The first major issue of this study is the authors' choice and lack of clear motivation to compare the SEF which occurs *after RI*, with the intensity fluctuation that occurs *during the RI period* (i.e., section 3.3.1 to 3.3.4). These two types of events occur at different stages of the TC lifecycle (*during* versus *after* RI). The simulated SEF events occur near T+77h, with T here denoting the model initialization time at 05 Sept 1200UTC, which implies that the SEF roughly occurs near 08 Sept 1700UTC. On the other hand, the first episodes of simulated intensity fluctuation occur near T+44, with T here representing the model initialization at 03 Sept 0000UTC, which implies that the intensity fluctuation occurs near 04 Sept 200UTC. This means that these two events have a time difference of almost 3.5-4 days (about 93 hours). Although the authors did not show intensity time series in this manuscript, from their recently published Torgerson et al. 2023 and it is shown that the corresponding intensities of these two types of events are significantly different (i.e., 950 hPa for intensity fluctuation and near 910 hPa for SEF). This important information is not mentioned anywhere in this manuscript.

We agree that the short-term intensity fluctuations occur during RI, while an ERC happens at the end of RI. In the later case this is because the reorganisation is disruptive enough to terminate the RI. We do understand that given this large disparity in consequences of the respective fluctuations it may seem arbitrary to compare them, however, the main motivation for doing so is that we believe they are governed, especially initially, by the same dynamical processes which makes the divergence in outcome less obvious than the reviewer suggests.

It should be noted additionally that the difference in starting intensity only applies to the W1 intensity fluctuation. There were four short-term intensity fluctuations that were similar, even though we focused on W1 in this paper and Torgerson et al (2023), W4 which occurred more than two days later behaved qualitatively similar to W1. Further information is shown about W4 aswell as the other strengthening and weakening phases is available in Fig.4 of Torgerson et al (2023). Additionally, the pressure during W4 started at around 918 hPa so was similar in overall intensity to the EWRC. This can be seen in the Figure R3, the minimum sea level pressure at the end of S3 is almost identical to the minimum sea level pressure at the end of stage c of the SEF.

[Figure]

*Figure R3: Minimum sea level pressure for both simulations with intensity fluctuations from the 0300 UTC simulation (black line) labelled and the stages of the ERC labelled for the 0512 UTC (purple line) simulation. The stages are the same as shown in the plot in Fig. 8. in the original manuscript.*

Because of the large difference in time and intensity when the two types of events occur, many aspects of the storm are drastically different, such as inner-core wind structure, inertial and symmetric stabilities, relative humidity distribution, and very importantly the environment in which the storm was embedded (see my comment 4 below). It is only until very late in the manuscript that the authors showed the $\theta e$ cross sections in Fig. 13, which gives the readers some senses of how different the TC vortex structure is between those two events. As in Fig. 13a-d, after RI (when SEF occurs), the $\theta e$ at the TC inner core is significantly higher, with contours that are more vertically oriented. This $\theta e$ structure indicates a more symmetrically neutral eyewall structure and must be associated with a significantly more intense tangential wind structure and inertial stability (which is not shown!), as well as less tendency for the inner eyewall to further intensify. In contrast, during RI, the TC intensity must still be below its potential maximum intensity and the primary eyewall has more conditional/symmetric instability. As shown in Fig. 13e-h, the eyewall $\theta e$ during the early intensity fluctuation has a clear inward bending structure and decreases vertically near z = 2 - 5 km, indicating the eyewall has a greater conditional/symmetric instability and thus greater tendency to further intensify. However, the TC intensity and basic state vortex structure are not mentioned or examined (not in the discussion of Fig. 13, nor presented in any of the earlier figures), nor considered to be relevant factors contributing to the distinct behaviors in the intensity fluctuation.

We admit that the base state of the TC should have been explicitly pointed out. However, the purpose of Fig. 13is to show how the thermodynamic structure changes during the fluctuations; in particular highlighting how there is a quasi-irreversible change that occurs in the ERC that does not occur in the short-term intensity fluctuation. Figure R4 below compares the same EWRC with W4 from the 0300 UTC simulation. Despite the more intensified base state a similar process occurs with the radial gradient of equivalent potential temperature being able to recover. We have expanded the module evaluation section to include discussions about the base state of the storm and environmental conditions. We have not included this additional figure in the manuscript as we feel it does not add any additional information beyond what is already shown. We have investigated

inertial stability for this storm but did not find any significant trend beyond $T_1+30h$ and do not think the diagnostic adds significant value to the study.

[Figure]

*Figure R4: Equivalent potential temperature (K, shaded), vertical velocity (0.1, 0.2, 0.5, 2, 5, 10 ms−1 positive (solid) and negative (dashed). Panels (a-d) from ensemble member 10 of the 05 September 12 UTC simulation showing the full ERC and panels (e-h) for ensemble member 15 of the 03 September 00 UTC simulation showing the intensity fluctuations. Also shown is the height dependent RMW (dashed grey line).*

Fundamentally, given the large differences in the basic-state vortex structure and intensity, *why should we expect that the TC would undergo a similar evolution? In other words, what makes the authors think that these two types of events are comparable to one another?* To be honest, the entire section 3.3 reads like it is comparing apples with oranges and *then concludes that apples and oranges are different*.

Respectfully the entirety of section 3.3 does not emphasise the differences between the types of fluctuations. Rather, the majority of section 3.3 focuses on the similarities between the two fluctuations and essentially posits that the short-term intensity fluctuations are governed by similar unbalanced boundary layer feedback mechanisms occurring in different contexts (outer rainband activity compared to inner rainband activity). Thus it is overriding similarities in disparate contexts that make the comparison interesting and useful. The eyewall demise section (3.3.4) does focus on a significant difference and explains why, *despite*, similar initiating mechanisms the end result of the fluctuations is radically different. To use your analogy the paper argues that while apples and oranges are different, they are still both fruit even though there is no reason to intuitively think that they can be related. We admit this was not clear in our original paper so have expanded the introduction of section 3.3 to make this point more easily understood.

For example, near L254-268 where the authors discussed Fig. 9, they compared the low-level PV evolutions between SEF and intensity fluctuation. The main findings (summarized below) are that

- The increase in azimuthal symmetry in the intensity fluctuation was attributed to the weakening of wave–2 inner rainband structures, but not present in the ERC.

 - The changes in azimuthal PV symmetry during the ERC are smaller than the intensity fluctuations.

- The maximum standard deviation of PV does not change much during ERC, whereas for the intensity fluctuation, there is a rapid decrease in the standard deviation of PV during a weakening phase and an increase in the standard deviation of PV during a strengthening phase.

But what are the respective implications of these three findings? Given that in the SEF case, we know that there is an outer eyewall forming, which then undergoes ERC, whereas in the intensity fluctuation, there is no outer eyewall formation, all we can conclude here is that they evolve differently because they are fundamentally different processes that occur in different background vortex state. In fact, the authors also did not provide any further discussion about the implications of these differences.

We agree we could have done a better job motivating the reasons behind the analysis and the comparisons with the two types of fluctuations. The reason for these diagnostics (which was touched on in the conclusions of the original paper) is that it is proposed that the initiating mechanism for both fluctuations is similar in terms of balanced adjustments and suction effects from inner or outer rainbands (potentially aided by VRW activity) giving rise to an unbalanced positive feedback within the boundary layer that leads to a major

structural alignment that materializes as a SEF in the case of the ERC and a reorganisation of the eyewall in the case of the fluctuations. We have better motivated this analysis by adding descriptions to this section about how the results relate to the mechanism involving a collapse of a barotropically unstable state in terms of the short-term intensity fluctuations.

Similar issues go with the subsequent sections. For instance, in L280-283 describing intensity fluctuation, "*However, unlike in the case of the eyewall replacement cycle, there is no updraught (associated with outer rainbands) outside of the eyewall and above the boundary layer that moves radially inwards over time before merging with the boundary layer updraught associated with the newly forming secondary eyewall.*" This is essentially saying that the intensity fluctuation is different from SEF because there is no mechanism to form a new eyewall. But, *do we need to compare SEF with intensity fluctuation in order to learn about the role of the developing rainband in the SEF process?* The answer is no because the importance of outer rainband development has been identified in many other studies (see my major comment 3), all of which did not compare SEF with intensity evolution to reach this conclusion. *So, what are we expecting to learn by comparing SEF with intensity fluctuation?*

The outer rainband is not involved in the case of the short-term intensity fluctuations but a proposal is that the inner rainband could play a similar role. And while we suggest there are similarities between the two fluctuations, the radial difference does seem to matter; in this case with a large separation between the inner and outer pillars of convection in the case of the SEF. We have added how this is relevant to the overall story at the end of this same section.

Most of the discussion in sections 3.3.1 to 3.3.4 simply highlights the *intrinsic differences* between the SEF and intensity fluctuation, all of which are not surprising given they occur in substantially different intensities and vortex states and are undergoing distinct evolutions. The fundamental issues that need to be addressed here are *what are the motivations to do these comparisons*?

As noted above, our fundamental motivation is to understand whether the mechanisms involved with the two fluctuations could be similar albeit with inner rainbands rather than outer rainbands in the case of the SEF and, if this is the case, why are the evolutions different? Extensive modifications to the discussion section was undertaken, including a detailed discussion of the difference between the mechanisms in the respective fluctuations.

*What are the justifications to convince the readers that these two types of events are comparable to one another, so that it is possible to identify a nontrivial cause leading to the observed differences?*

Both fluctuations involve convection, outside of the eyewall, that disrupt the structure of the storm (albeit to different extents), with a similar unbalanced boundary layer feedback mechanism involving an increased radial extent of agradient wind and region of poor ventilated convection.

And finally, *what meaningful conclusions can we draw based on the observed differences?*

The radial separation between the eyewalls in the case of the SEF is significant and leads to a much more drawn out and extreme structural realignment compared to what happens when inner rainbands are axisymetrisized. We have elaborated on these points, particularly in the discussion section.

**2. Confusing time referencing** As I mentioned in major comment 1, it takes the reader quite some effort to compare different figures in the manuscript to realize the exact time difference between these events. Many of the Hovmoller diagrams start with some positive hours, such as 70 h in Fig. 2, 3, and 60 in Fig. 6. I presume for SEF-focused figures, these times are counted from the *initialization time at 05 Sept 1200UTC*. On the other hand, for intensity fluctuation-focused figures, such as Fig. 8b, 9e-h, 10a-h, and 11a-h, the times T+xh are counted from *the initialization time at 03 Sept 0000UTC*. This time labeling scheme is misleading, because, for example, it gives the impression that these two events are about 33 hours apart (comparing Figs 9a-d and 9e-h). However, because of the initialization time difference, there is an additional 60 hours of time difference (i.e. time difference between 03 Sept 0000UTC and 05 Sept 1200UTC), making the total time difference of 93 hours time difference, which is almost 4 days. This approach of time referencing hides the large time difference between the two events, which is inappropriate. Therefore, the authors need to come up with a new time referencing approach that accurately describes the actual time difference between the two types of events. One possible approach is to use different labels in all figures that currently use the label $T$, such as $T_{0309\_00Z}$ and $T_{0509\_12Z}$ to represent the two initialization times.

We have relabelled all the figures to explicitly refer to the reference time of either the 03 September 00UTC initialized run (run 1) or the 05 September 1200UTC run (run 2).

**3. SEF mechanism related to outer rainband dynamics** This study only examined two VRW-related hypotheses (i.e., VRW stagnation radius and filamentation) and boundary layer unbalanced processes. However, there is a large body of studies emphasizing the rainband-driven SEF mechanism. Here is listed just a small portion of them: Qiu and Tan 2013; Li et al. 2014; Zhu and Zhu 2014; Zhu et al. 2015; Tang et al. 2017; Chen 2018; Chen et al. 2018; Yu et al 2021a,b, 2022. The only study that the authors cited in the introduction is Didlake et al. 2018. But rainband-related SEF hypothesis was proposed as early as Didlake and Houze 2013 and has since been further developed by many subsequent modeling and observational studies.

We do not dispute the importance of the rainband forcing, though we may not have been clear about it in our paper. Indeed both types of fluctuations rely on rainband forcing which forms part of the basis for comparing them. For example, in the Zhu and Zhu (2014) study a Sawyer Eliassen approach demonstrates the balanced effect of the outer rainband activity which leads to the broadening of the tangential wind field through balanced adjustment

which we have proposed as an important element to enhance radial outflow above the boundary layer in both types of fluctuations. In a sense rainband forcing is viewed almost as an axiom with this paper focusing more on the processes which convert asymmetric rainband activity into ring convection (in the case of SEF) or a disrupted, bifurcated eyewall in the case of the short-term intensity fluctuations. We have expanded the start of section 3.2 to emphasise these points.

For instance, Qiu and Tan (2013) examined the connection between unbalanced boundary layer response to asymmetric inflow induced by the outer rainband. This asymmetric inflow is also similarly identified in observation (Didlake and Houze 2013; Didlake et al. 2018) and many modeling studies (Dai et al. 2017; Zhang et al. 2017; Chen 2018; Chen et al. 2018; Yu et al. 2021a,b and 2022), and is referred to as "mesoscale descending inflow" (Didlake and Houze 2013; Didlake et al. 2018; Yu et al. 2021a and 2022). Yu et al. 2021a and 2022 demonstrated the important role of a mesoscale descending inflow within the stratiform precipitation region in initiating the broadscale wind field acceleration and boundary layer cold pool dynamics in sustaining convective updraft at the inner edge of the descending inflow. From observation, it has even been demonstrated that 79% of observed SEF events have a stationary rainband complex within 6 hours of the SEF development (Vaughan et al. 2020). Given all these important rainband-focused SEF research and the apparent importance of rainband processes in the present case, I am surprised that the author neglected this large body of literature and only focused on examining the few hypotheses that do not emphasize rainband dynamics.

We do agree that the literature section may be deficient to not talk about the role of rainbands more explicitly and in depth, however, it is not out of a lack of appreciation of their importance. We accept that rainband forcing, including asymmetric effects, is import; indeed the VRW and beta skirt mechanisms discussed are inherently asymmetric in nature and intrinsically linked to the inner and outer rainbands for the two kinds of fluctuations. For the case of the short-term intensity fluctuations we devoted significant time to the effect of the wave-2 asymmetry in Torgerson et al 2023. In this current research we do provide some axisymmetric analysis, but we never meant to imply that the storm is symmetric, merely that many of the important effects of the rainband activity can still be described from their effects (balanced or otherwise) projected onto the axisymmetric state. Additional studies have been added to the introduction and the role of the outer rainband has also been made clearer in the context of this study. We have also added stronger motivation at the end of the introduction section that emphasises the role of both the outer and inner rainband in the proposed mechanism.

Here, I want to clarify that I do agree that the unbalanced boundary layer process is an important part of the SEF mechanism. However, it should be noted that the unbalanced boundary layer process argument does not emphasize the importance of the rainband process *as a precursor of SEF*. Given that in the present case, the rainband development precedes the onset of SEF (or as the precursor of SEF), it is unreasonable not to examine whether the simulated SEF event shares similarities with the hypotheses and findings from the other rainband-related SEF studies. Therefore, the entire section 3.2 seems to be incomplete to me.

We did emphasise the broadening of the wind field which is a direct consequence of the rainband activity and is still the case even if the VRW or beta skirt mechanisms play a role. We would expect it to share similarities with the other rainband studies. We should emphasise we do not dispute the important roles of the rainbands, the focus of this study is the period after the rainband has already become established and tangential wind broadening has begun, or in the case of the short-term intensity fluctuations the weakening phase has started. We are now much more explicit about this at the start of section 3.2.

**4. Environmental wind shear is the largest organizing factor of the TC rainband development** Given the importance of rainband development in the simulated SEF case, it is important to show the environmental controlling factors related to rainband development. Specifically, it is well known that the largest organizing factor of rainbands is environmental shear. However, this study mentions nothing at all about the role of environmental shear. How does the environmental shear vary over time, particularly during the two periods of intensity fluctuation events and SEF? Are there any differences in shear magnitude and direction in shear during these two types of events? Based on a quick search of relevant publications, Fischer et al. 2020 showed that the 850-200 shear increases to about 5 $ms{-1}$ between 07-08 Sept during the SEF event, while near 04-05 Sept the VWS was only about 2-3 $ms{-1}$ (see their Fig. 3a). This increase in VWS is likely an important environmental controlling factor that contributes to the development of outer rainband in the SEF event (both actual or simulated). This also explains why the rainband during RI is much less developed due to weak shear in the environment. Again, this important piece is entirely missing in this study.

[Figure]

*Figure R6: 850–200mb vertical wind shear in 3 to 5 degree annulus from model TC centre as a function of time. Thresholds defining weak, moderate, high or extreme (where TC development is impossible) are also based on Park et al. (2012).*

We acknowledge that wind shear is a key environmental factor and chose Irma as a case study because environmental conditions were reasonably conductive to development

throughout. Figure R5 shows the modelled shear for both forecasts along with the intensity fluctuation and ERC period labelled. Throughout the period modelled by at least one forecast from 03 September 00UTC to 09 September 12UTC shear is predominantly weak or moderate and only occasionally high briefly on 08 September (it should be noted that this calculation likely overestimates vertical shear due to the annulus intersecting with the Hispaniola island during this time). During the intensity fluctuations the modelled vertical shear varies from a weak 1ms-1 at the start of S3 to a moderate 6ms-1 at the start of W1 with the wind shear mostly being weak throughout the intensity fluctuations. During the SEF event the wind shear is a borderline moderate 4-6ms-1 although is a bit higher prior to the SEF and during the replacement. Given the similar vertical shears during both types of fluctuations (noting the effect of Hispaniola island) we do not believe either fluctuation was linked to changes in wind shear especially since we could find no link between ensemble forecast storm track and the probability of either fluctuation occurring. It should be worth noting that the storm during the intensity fluctuation period did not lack an outer rainband, simply that SEF did not occur. These points have been summarised in the model evaluation section of the results.

**5. Emergence mechanism of supergradient wind** While I agree that the unbalanced boundary layer process is an important part of the SEF mechanism, one major issue of the argument is that the boundary layer spin-up mechanism does not explain what exactly drives the emergence of supergradient wind. As stated in L388, the authors do not seem to delve into the detailed analysis to determine what causes the enhanced inflow, but merely point to the positive feedback between rainband and boundary layer processes, i.e., expanded tangential wind field, enhanced boundary layer inflow and the subsequent emergence of supergradient wind and updraft? I would say this feedback mechanism is not new. If the authors are reluctant to determine the cause of enhanced boundary layer inflow, then what is exactly new here?

The unbalanced feedback mechanism as a whole is not new (though the aspect involving outflow enhancement through poor moat ventilation is very recent) but the link between, as you point out, two very different types of intensity fluctuations both being explained by a similar initiating process is novel and has not been done before. Therefore, the key result of the paper is that this same process applied to two different types of rainbands results in fluctuations with different characteristics. We have elaborated on these points in the first part of the discussion section.

**6. Other causes of intensity fluctuation** For pre-intensifying TC in shear, intensity fluctuation is often caused by the inward intrusion of low-$\theta_e$ air into TC's inner core, a process known as ventilation. This ventilation is not the same as the mass ventilation out of

the boundary layer (diagnosed in section 3.3.3) that relates to the deepening of surface pressure. Low-$\theta_e$ ventilation is a well-known factor that can cause the weakening of TC (Tang and Emanuel, 2010, 2012a,b) or lead to delay of intensification, specifically for TCs in shear that are before reaching its mature state. Specifically, it is known that there exist several pathways of ventilation, namely the mid-level radial ventilation and the low-level downdraft ventilation. Alland et al. 2020a,b showed a nice summary of these pathways. A recently published study (Yu et al. 2023) also show how low-$\theta_e$ ventilation into TC inner core can cause substantial weakening of TC intensity. These possible mechanisms of intensity weakening are not examined or mentioned in the manuscript.

For our study of the short-term intensity fluctuations (see Torgerson et al. 2023) we specifically chose a storm that was under low vertical shear in an attempt to limit the impact of such mechanisms. We have also shown (see Fig. 13 of original manuscript) that the intrusion of low $\theta_e$ air into the eye  happens in the ERC which leads to the contributes of the inner eyewall. There is no analogous process in the short-term intensity fluctuations . In terms of the purpose of section 3.3.4 which looks at the role of the eyewall demise from a thermodynamic perspective, the main motivation was looking at why the short term intensity fluctuations did not see a destruction of the eyewall, the mechanisms highlighted here would not apply to the short term fluctuations which does add weight to our proposition that the key difference is the radial separation between the eyewall and the secondary eyewall.  We have included additional consideration of the vertical wind shear in the discussion section related to the eyewall demise.

Overall suggestions to resolve these major issues and improve the manuscript: Based on the major issue discussed above, I strongly suggest the authors rethink the necessity of comparing SEF events with intensity fluctuation that occurs in a much weaker intensity state. At this point, this manuscript is structured in a way that the comparison is not necessary. If this paper aims to focus on examining the SEF process, I strongly suggest the author look into the large body of rainband-related SEF studies and delve into the rainband structure to examine the linkage and feedback between TC rainband and unbalanced boundary layer processes. If this paper aims to focus on intensity fluctuation, then I also agree with Dr. John Methven's comments about the need to distinguish the current paper from the authors' recently published 2023 paper.

We agree we need to make the aims and motivations of the paper clearer. Our principal motivation was to highlight that the two types of intensity fluctuations share a common proposed initiation mechanism (as described in Torgerson et al. 2023) in terms of a dynamical adjustment to convection outside the eyewall (albeit inner and outer rainbands). Therefore, the focus of this manuscript is to analyse the unbalanced feedback mechanism that occurs in both types of fluctuations and to determine why the end result plays out so differently. We highlight the importance of the separation between the SEF region and the primary eyewall in the case of the EWRC.

**Minor comments:**

Title: The title is incorrect. The eyewall replacement cycle does not happen during a period of

rapid intensification period, but after the rapid intensification.

The title has been changed to 'Comparing short term intensity fluctuations and an Eyewall replacement cycle in Hurricane Irma (2017) during and immediately after a period of rapid intensification'.

L11: Nascent TC is susceptible to environmental shear, which could cause a substantial delay in intensification. During this delay, intensity can remain nearly steady as the TC vortex undergoes precession. So the statement that the intensity of newly formed TC typically increases over time is incorrect.

Clarified this initial statement.

L38: outflow jet. Jet usually is used to describe localized enhanced wind fields. For TC vortex, the strongest wind is mostly likely along the tangential direction, e.g., supergradient jet.

Fixed by referring to an 'outflow layer'.

L41-46: This paragraph aims to provide the motivation, but is weak. Only stating that this study aims to compare these two events is not sufficient. As explained by my major comment 1, the background state TC vortex is drastically different, making a direct comparison of these two events nearly impossible to extract meaningful conclusions, other than saying that they are intrinsically different processes that evolve entirely differently. This paper needs to substantially strengthen the motivations by providing stronger reasoning, such as explaining why the authors think that these two processes are comparable. Why is it necessary to compare these two processes?

The section has been expanded including the addition of the importance of the rainband activity.

Introduction: Overall, the discussion about rainband dynamics in the introduction is not enough. There is only one sentence in L25-27 citing Didlake et al 2018. Given that rainband dynamics plays a clear role in the simulated SEF event, the authors need to provide a brief survey about rainband dynamics in a sheared environment to allow readers background information about the role of rainbands in the SEF process, a summary of what previous rainband-related SEF studies had learned (see major comment 3), and what still remains unclear.

L60: that's brightness is inversely correlated to → which has brightness inversely correlated to

Fixed.

L99: The SEF event happened earlier in reality → The observed SEF happened earlier in reality.

Fixed.

Section 3.1: If this study aims to compare the SEF with intensity fluctuation, this section should not only focus on the SEF event but also provide basic information about the vortex structure during intensity fluctuation. Specifically, a comprehensive comparison of the vortex structure (e.g., tangential wind, moisture) when these two events happen is also necessary to show readers the basic vortex structures when the two events happen and how comparable they are. Also, discussions about the difference in intensity (showing intensity time series) and environmental wind shear (also time series) are necessary.

We have split section 3.1 into two sections relating to the eyewall replacement cycle and the short-term fluctuations in addition to a preamble discussing the storm structure during the entire period of interest.

.Section 3.1.1 gives more information on the vortex structure during the short-term intensity fluctuations and references our previous paper where time series (Fig. 4) are shown for tangential wind speed, radius of maximum wind speed and minimum sea level pressure. Fig.R3 and Fig. R6 show time series of the intensity and environmental shear respectively. We would highlight that the intensity fluctuations occur over a wide range of intensities up to and including a similar intensity where the SEF occurred, and that the vertical shear was similar in both types of fluctuations. We have not elected to include these figures in our study because their main purpose is the elimination of environmental conditions playing a large role compared to internal or stochastic processes.

Section 3.2: This list of SEF mechanisms is not comprehensive. SEF-related mechanisms also need to be examined. See major comment 3.

We do admit this was unclear, we consider the outer rainband to play a pivotal role, however the focus of the study is not on the formation of the outer rainband but the transformation of the outer rainband to a secondary eyewall. Thus our mechanisms relate to this transformation and all include the outer rainband as a necessary criterion including the dynamical response of the rainband which marries synergistically with the mechanisms we have described. We have clarified this in our section 3.2 preamble.

L259-260: A theory of barotropic instability across the moat in the ERC process (Lai et al. 2019; Lai et al 2021a,b) has been proposed in recent years that explain the elliptic shape of the PV structure during ERC, as well as the rapid decay of the inner eyewall, which are clearly relevant to this study.

In Torgerson et al (2023) we talked about this concept in the context of the short-term intensity fluctuations where barotropic instability seemed to increase in strengthening phase up until the point the diabatic heating could no longer maintain the unstable PV distribution and the barotropic instability fell. One result in this new study is that something similar did not occur in the case of the secondary eyewall formation event until the time when the replacement started indicating a greater delay between the weakening and decrease in barotropic instability. We have added this explanation at the start of section

3.3.1 and also a brief discussion of the results of Lai et al and type 2 barotropic instability from Kossin et al 2000 later in the same section.

Figures: Many of the plan-view figures do not have spatial markers in the x and y axes. Even though the authors added circles representing 25 and 50 km radii, but showing markers in the axes can let the readers see the exact size more clearly. Also, the black circles overlap with the black contours, making them difficult to see (e.g, Fig. 9). Also, in Fig. 1, there is no circle nor markers in the axes, which are absolutely needed. How can we tell if the simulated TC size is realistic compare to the observation?

We have made changes to the figures to accommodate these requests.